

# Sources of reactive nitrogen in marine aerosol over the Northwest Pacific Ocean in spring

Li Luo[1,3], Shuh-Ji Kao[2]*, Hongyan Bao[2], Huayun Xiao[1,3], Hongwei Xiao[1,3], Xiaohong Yao[4], Huiwang Gao[4], Jiawei Li[5], Yangyang Lu[2]

[1]Jiangxi Province Key Laboratory of the causes and control of Atmospheric pollution, East China University of Technology, Nanchang 330013, China
[2]State Key Laboratory of Marine Environmental Science, Xiamen University, Xiamen 361102, China
[3]School of Water Resources and Environmental Engineering, East China University of Technology, Nanchang 330013, China
[4]Key laboratory of Marine Environmental Science and Ecology, Ministry of Education, Ocean University of China, Qingdao 266100, China
[5]Key Laboratory of Regional Climate-Environment for Temperate East Asia, Institute of Atmospheric Physics, Chinese Academy of Sciences, Beijing 100029, China

*Correspondence to*: Shuh-Ji Kao (sjkao@xmu.edu.cn)

**Abstract** Atmospheric deposition of long range transport anthropogenic reactive nitrogen (Nr, mainly $NH_x$, $NO_y$ and water-soluble organic nitrogen (WSON)) from continent exerts profound impact on marine biogeochemistry. On the other hand, marine biogenic dissolve organic nitrogen (DON) is also a potential contributor to aerosol WSON in overlying atmosphere. Despite of the importance of off-continent dispersion and interactive processes of Nr in the atmosphere-ocean boundary, knowledge regarding sources of various nitrogen species in the open ocean remained limited due to insufficient observations. In spring of 2014 and 2015, we conducted two cruises starting from the coast of China through the East China Seas (ECSs, i.e. Yellow Sea and East China Sea) to the open ocean (i.e. Northwest Pacific Ocean, NWPO). Concentrations of water-soluble total nitrogen (WSTN), $NO_3^-$ and $NH_4^+$, and $\delta^{15}N$ of WSTN and $NO_3^-$ in marine aerosol were measured for both cruises. In spring of 2015, we also analysed the aerosol CHON molecular formulas by using Fourier transform ion cyclotron resonance mass spectrometry (FT-ICR-MS), and the concentrations and $\delta^{15}N$ of $NO_3^-$ and DON of surface sea water (SSW) (depth 5m) along the cruise track. Aerosol $NO_3^-$, $NH_4^+$ and WSON showed a logarithmic off shore decrease pattern (1-2 orders of magnitude drop) reflecting a strong anthropogenic emission source of $NO_3^-$, $NH_4^+$ and WSON from China. Concentrations of aerosol $NO_3^-$ and $NH_4^+$ were significantly higher in 2014 (even in remote NWOP) than those in 2015 due to stronger wind field in 2014, underscoring the role of the Asian winter monsoon in seaward transport of anthropogenic $NO_3^-$ and $NH_4^+$. However, the WSON in background aerosol over the NWPO in 2015 (12.7±8.7nmol m$^{-3}$) was close to that in 2014 (10.7±7.0nmol m$^{-3}$) suggesting an addition of non-anthropogenic WSON source in the open ocean. Results of the FT-ICR-MS analyses revealed a higher contribution of marine biological source to WSON in background aerosol and this was supported by the analysis of isotopic mixing between aerosol $NH_4^+$ and SSW DON (5.8±2.0‰). Obviously, the marine DON emission should be considered in model and field work while assessing the net atmospheric WSON deposition in the open ocean. This is the first hand parallel dataset for isotopic compositions of marine DON and



aerosol Nr, more studies are required to explore the complicated processes of sources and deposition of Nr to advance our knowledge of the anthropogenic influence on marine nitrogen cycle and nitrogen exchanges through the land-ocean and the air-sea interface.

# 1 Introduction

Atmospheric transport and deposition of anthropogenic reactive nitrogen (Nr) to global ocean has increased profoundly since the industrial revolution (Duce et al., 2008). Due to cumulative atmospheric Nr deposition, the stoichiometric

relationship between nitrogen and phosphorous (nitrogen is the limiting nutrient in most surface oceans) in upper ocean has been significantly altered at least in the North Pacific (Kim et al., 2011). Such alteration may impact oceanic pristine ecosystem and biogeochemical cycles as a consequence. The Nr deposited to the ocean includes inorganic reduced nitrogen forms ($NH_3$ and $NH_4^+$), oxidized nitrogen forms ($HNO_3$ and $NO_3^-$) and organic nitrogen compounds (Erisman et al., 2002) as well. The deposition fluxes (including both dry and wet) of atmospheric Nr to the global ocean have been studied previously

through models (Duce et al., 2008; Doney et al., 2010). Recent studies by model (Kanakidou et al., 2012) and observations (Alitier et al., 2014, 2016) also reported that ocean may be a potential source for atmospheric WSON and $NH_3$. Nevertheless, field observations in the open ocean are still insufficient, thus, multiple approaches such as molecular characteristics of WSON and stable nitrogen isotopic compositions are urgently needed to trace sources of Nr and to investigate the Nr exchange between the air-sea atmospheres.

Based on organic nitrogen compounds, Cape et al. (2011) concluded several possible sources of WSON in atmosphere including livestock and animal husbandry, fertilizers, vehicle exhaust, biomass burning, secondary pollutants and marine biological source. Cape et al. (2011) also pointed out explicitly that the complex atmosphric chemical process may obscure the source identification by individual organic N compounds for atmospheric WSON. Nevertheless, stochastic analysis coupled with the molecular characterization approach by using FT-ICR-MS reveled biological organic nitrogen in the SSW

can be a source of atmospheric WSON in the open ocean (Wozniak et al., 2014; Altieri et al., 2016). Similar conclusion was made according to positive correlations between marine aerosol WSON concentration and wind speed in a cruise to the Northwest Pacific ocean (Luo et al., 2016). Here in this study, we also tried using FT-ICR-MS to characterize organic nitrogen compounds in marine aerosol.

Stable nitrogen isotopic compositions could be a potential tool to discriminate the sources of atmospheric $NO_x$ and $NH_x$ by

their isotope values. This approach ($\delta^{15}N$-$NO_x$) has been successfully applied to distinguish $NO_x$ of fossil fuel burning from soil biogenic activity (Felix and Elliott, 2014), as well as between coal combustion (Felix et al., 2012) and vehicle exhaust (Walters et al., 2015). Similarly, atmospheric $NH_x$ also can be measured and traced by using $\delta^{15}N$-$NH_x$ (Freyer, 1978; Heaton, 1987; Jickells, 2003; Altieri et al., 2014). However, direct measure of atmospheric $\delta^{15}N$-WSON is almost impossible presently due to incomplete separation between inorganic and organic nitrogen. Via isotope mass conservation, a few

previous studies reported values of $\delta^{15}N$-WSON in precipitation collected from urban, rural and remote regions with a range





from -7.3 to +7.3‰ (Cornell et al., 1995), which was consistent with those in precipitation sampled in a metropolis surrounded by agricultural areas in southern Korea (-7.9 to +3.8‰, with annual mean of +0.3‰ and +0.2‰ in 2007 and 2008, respectively; Lee et al., 2012), but lower than these $\delta^{15}$N-WSON (-0.5 ~ +14.7‰, with median of +5‰ and insignificant seasonal variation) reported in precipitation in the US east coast area (Russell et al., 1998). Compared with

precipitation, wider range (mainly caused by low values) of aerosol $\delta^{15}$N-WSON was reported in various rural regions in the UK (-14.6 ~ +12.5‰, with median of -2‰ and -5‰ for fine and coarse mode; Kelly et al., 2005)( Fig. S1). Compared with $NO_x$ and $NH_x$, it is more difficult to identify the sources of atmospheric WSON by using $\delta^{15}$N. However, the relatively uniform $\delta^{15}$N values (+2.2 to +5.4‰) of dissolved organic nitrogen (DON) globally in SSW (Knapp et al., 2005; Knapp et al., 2011) shed lights on isotope end member mixing approach for marine aerosol WSON, unfortunately, no parallel sampling

during a cruise for marine aerosol and SSW $\delta^{15}$N identification so far.

In terms of hemispheric wind field, the influence of the East Asian monsoon transition from winter (from October to April) to summer (from May to September) covers the whole East Asian region. During the period of East Asian winter monsoon, strong cold air mass mobilizes rapidly through the northeast of China to the NWPO; by contrast, the air masses mainly sourced from the tropical Pacific Ocean during the summer monsoon (Wang et al., 2003). It have been reported that

the air masses originated from the China in winter contained higher concentrations of $NO_x$ and $NH_x$ than air mass sourced from the remote Pacific in summer in oceanic regions (Kunwar et al., 2014). Due to variable monsoon intensity, monitoring over the NWPO at the Hedo and Ogawasara islands also showed that the dry deposition of aerosol $NO_3^-$ and $NH_4^+$ varied inter-annually by a factor of 2-5 (http://www.eanet.asia/). Meanwhile, dust storm frequently outbreaks during transition period and long-range transport in the upper and mid troposphere through north of China to the remote Pacific (Yang et al.,

2013), and the dust plume contains abundant crustal elements additional to $NO_x$ and $NH_x$ (Duce et al., 1980; Kang et al., 2009). To evaluate the seaward gradient of atmospheric Nr concentration and to explore the source and fate of atmospheric $NO_3^-$, $NH_4^+$ and WSON from the largest emission country, China, we select spring, the transitional period of East Asian monsoon, to conduct cruises to the Northwest Pacific Ocean. Results from two different years allowed comparison to be done.

In this study, we measured the concentrations of WSTN, $NO_3^-$, $NH_4^+$, and $\delta^{15}$N-WSTN and $\delta^{15}$N-$NO_3^-$ in marine aerosols collected cover the ECSs and the NWPO in spring of 2014 and 2015. The concentrations and $\delta^{15}$N of DON and $NO_3^-$ in SSW (depth 5m) were analysed in parallel along the cruise track in 2015. The molecular characteristics of water soluble CHON (composed of C, H, O and N) molecular formulas were also analyzed by the FT-ICR-MS for the 2015 aerosol. The purposes of this study are (1) to investigate the spatial distribution of concentrations of various Nr species in marine

aerosol from the ECSs to the NWPO, (2) to explore possible sources of atmospheric WSON in marine realm, finally, to advance our knowledge about the atmospheric Nr transport at land ocean boundary and the potential Nr exchange between atmosphere and ocean.





## 2 Material and Methods

### 2.1 Sampling and background weather during cruises

Marine aerosol samples were collected during two research cruises (Fig. 1) by using the R/V *Dongfanghong* II. The first cruise (Fig. 1a) was from 17 March to 22 April in 2014 (total 44 samples were collected and sampling information can be found in Luo et al., 2016). The second cruise (Fig. 1b) was from 30 March to 4 May in 2015 (total 39 samples, detailed sampling information including date, time period and locations for each sample are listed in Table S1). Both cruises were implemented during the transition period of the East Asian monsoon. The 2-months (March and April) average wind

streamlines at 1000 hPa over the NWPO showed the ranges of wind speed from 2 to 6 m s$^{-1}$ in 2014 (Fig. 1a) and from 1 to 3 m s$^{-1}$ in 2015 (Fig. 1b). In general, the wind was stronger in 2014 relative to that in 2015 during the sampling periods.

  Meteorological data, including wind speed and direction and relative humidity (RH) and ambient temperature, for 2015 cruise was shown in Fig. S2 (2014 cruise had been reported in Luo et al. 2016). Both cruises encountered sea fog in the ECSs, thus, the aerosol sampling and of course the aerosol chemistry was influenced by sea fog inevitably. According to

analogous weather conditions experienced in both cruises, we followed Luo et al. (2016) to classify 2015 marine aerosol samples into three types (i.e. sea-fog-modified aerosol (orange triangles) collected in the ECSs, dust aerosol (pink triangles) and background aerosol (blue triangles) sampled in the NWPO in Figure 1) basing on the meteorological conditions (Fig. S2), concentrations of aluminum (the data not show) and the lidar browse images from the NASA (Fig. S3).

  To examine the relationship of isotopic compositions between WSON in marine aerosol and DON in SSW, we collected

SSW (depth 5m, sampling locations are shown in Fig. 1b open black circles) by Niskin bottle during 2015 cruise. The SSW were filtered by 0.2 μm filter and kept frozen at -20 ℃ in 50mL 450 ℃ pre-combusted brown glass tubes until analysis.

### 2.2 Chemical analyses

### 2.2.1 $NO_3^-$ and $NH_4^+$ in marine aerosol

  All the marine aerosol samples were extracted by Milli-Q water (specific resistivity of 18.2 MΩ cm$^{-1}$) following Luo et

al. (2016). The aerosol extracts were analyzed by using an ion chromatograph (model ICS-1100 for anions and model ICS-900 for cations) equipped with a conductivity detector (ASRS-ULTRA) and suppressor (ASRS-300 for the ICS-1100 and CSRS-300 for the ICS-900). The precision for all ionic species was better than 5%. Details of the analytical processes can be found in Hsu et al. (2014). Only five out of all aerosol samples contained detectable $NO_2^-$ and accounted for < 1% of the WSTN.

### 2.2.2 $NO_3^-$ in SSW


  The concentration of $NO_3^-$ in SSW was measured by chemiluminescent method (Braman and Hendrix, 1989). Briefly, the solution containing $NO_3^-$ was injected into the heated solution of acidic Vanadium (III), in which $NO_3^-$ was reduced



into nitric oxide (NO) to be measured by NOx analyzer (MODEL T200U, Teledyne Technologies Incorporated, USA). Injection of working standards was carried out every 10 samples as an interval. The relative standard deviation for standard duplicates was < 5%. Concentration of $NO_2^-$ in SSW was below detection of 0.1 μmol L$^{-1}$ over entire cruise, as reported previously (Adornato et al., 2005).

### 2.2.3 WSTN in marine aerosol and total dissolve nitrogen in SSW

WSTN in aerosol and total dissolve nitrogen (TDN, i.e. $NO_3^-$ + $NH_4^+$ + DON) in SSW were measured by using the alkaline potassium persulfate oxidation method to convert WSTN and TDN to $NO_3^-$ (Luo et al., 2016; Knapp et al., 2005). The digested solution was then measured by chemiluminescent detection of the $NO_3^-$ (Braman and Hendrix, 1989). The recoveries of WSTN and TDN by the alkaline potassium persulfate digestion fell within 95 ~ 105% (n=6) over the range of detection.

### 2.2.4 Stable nitrogen isotope

The $\delta^{15}N$-$NO_3^-$ were analyzed by the denitrifier method described by Sigman et al. (2001) and Casciotti et al. (2002), which has been widely used to analyze the $\delta^{15}N$ values of $NO_3^-$ in aerosol, rainwater and sea water (Altieri et al., 2013; Buffam and McGlathery, 2003; Gobel et al., 2013; Hastings et al., 2003; Sigman et al., 2005), as well as $NO_3^-$ in digested solution by the alkaline potassium persulfate (Knapp et al., 2005; Knapp et al., 2010; Knapp et al., 2011; Knapp et al., 2012). The detailed analytical steps of stable nitrogen isotope can be found in Archana et al. (2016) and Yang et al. (2014). Briefly, $NO_3^-$ was reduced to N$_2$O by denitrifying bacteria, *Pseudomonas aureofaciens* (ATCC 13985), then the stable nitrogen isotope of N$_2$O was analyzed by using GasBench II connected to a continuous flow isotope ratio mass spectrometer (IRMS, Thermo Delta V Advantage). Two international standards USGS34 and IAEA-N3 (Böhlke et al., 2003) as well as two laboratory working nitrate standards were used to check the instrument stability. After WSTN and TDN were oxidized into $NO_3^-$, the $\delta^{15}N$-WSTN and $\delta^{15}N$-TDN were analyzed using the same procedures for $NO_3^-$. The pooled standard deviations for replicate were ± 0.2‰, ± 0.5‰ and ± 0.5‰, respectively, for $\delta^{15}N$-$NO_3^-$, $\delta^{15}N$-WSTN and $\delta^{15}N$-TDN.

### 2.2.5 Aerosol water soluble CHON molecular formulas

The water soluble CHON molecular formulas in aerosol were analyzed using a solariX FT-ICR-MS (Bruker Daltonic GmBH, Germany) in the negative ion mode and equipped with a 15 Tesla magnet at Oldenburg University. Briefly, half of the filter was extracted with Milli-Q water in an ultrasonic water bath, and filtered by 0.45μm nylon syringe filter. The extract was acidified to pH 2 with hydrochloric acid and concentrated by solid phase extraction (SPE) with styrene divinyl benzene polymer (PPL) cartridges. The PPL cartridge was rinsed using HPLC grade methanol. The extraction efficiency on a carbon basis was on average 46 ± 24% (mean ± SD, n = 44). Details regarding the FT-ICR-MS analysis are described in Bao et al. (2017).



### 2.3 data analysis

The concentrations of WSON in marine aerosol, which cannot be measured directly as aforementioned, were calculated
using the following equation:

$$[WSON] = [WSTN] - [NO_3^-] - [NH_4^+], \tag{1}$$

where $[WSTN]$, $[NO_3^-]$ and $[NH_4^+]$ are the molar concentrations (nmol N m$^{-3}$) of those water soluble nitrogen species in
marine aerosol.

The reduced nitrogen (RN, i.e. $NH_4^+$ + WSON) and its $\delta^{15}$N-RN in aerosol were calculated by mass balance:

$$[RN] = [WSTN] - [NO_3^-], \tag{2}$$

$$\delta^{15}\text{N-RN} = (\delta^{15}\text{N-WSTN} * [WSTN] - \delta^{15}\text{N-NO}_3^- * [NO_3^-]) / [RN], \tag{3}$$

where $[WSTN]$ and $[NO_3^-]$ are the molar concentrations (nmol N m$^{-3}$) of those water soluble nitrogen species in marine
aerosol. The average propagated standard errors for RN was 9% for both 2014 and 2015. The propagated error for the
calculation of $\delta^{15}$N-RN is $\pm 0.6$‰.

Similar to WSON in aerosol, the DON concentration and $\delta^{15}$N-DON in SSW were calculated using the following
equations:

$$[DON] = [TDN] - [NO_3^-], \tag{4}$$

$$\delta^{15}\text{N-DON} = (\delta^{15}\text{N-TDN} * [TDN] - \delta^{15}\text{N-NO}_3^- * [NO_3^-]) / [DON], \tag{5}$$

where $[TDN]$ and $[NO_3^-]$ are molar concentrations (μmol N L$^{-1}$) in SSW. The standard errors propagated through DON
calculation was 5.3%. Since $[NH_4^+]$ in SSW was typically less than 0.05 μmol L$^{-1}$ (Zhu et al., 2013), which is much less than
DON in μM level, thus, $[NH_4^+]$ is neglected in Equation 4 and 5. On the other hand, most of $NO_3^-$ concentrations in 5m
SSW of our case were always < 0.5μmol L$^{-1}$, which is too low for $\delta^{15}$N-NO$_3^-$ measurement. We tried to evaluate the
interference from nitrate for $\delta^{15}$N-DON calculation. The average occupation of $NO_3^-$ in total $NO_3^-$ plus DON was 5.7% for
all the samples in SSW, $\delta^{15}$N-NO$_3^-$ in SSW ranged from +8.2‰ to +16.4‰ in our measurements, the bias of calculated
$\delta^{15}$N-DON will be varied from +0.5‰ to +0.9‰.

The percent (P) of the primary marine biological source CHON formulas (CHON formula sourced from the sea spray
defined by the ratios of 0.15 < O/C < 0.45 and 1.5 < H/C < 2.0, the O/C and H/C ratios were calculated by dividing the
number of O (or H) atoms by the number of C atoms in an assigned formula; Wozniak et al., 2014) to the total CHON
formulas in a given sample was calculated using the following equations:

$$P = \sum F_{marine} / \sum F_{total}, \tag{6}$$

where $F_{marine}$ represents the intensity-weight of the CHON formula originated from the primary marine biological source and
$\sum F_{total}$ denote the intensity-weight of all assigned CHON formulas in a sample.

The N fluxes of dry deposition were calculated by:

$$F = C_i \times V_i, \tag{7}$$

where $C_i$ is for the aerosol concentration of water soluble nitrogen species and $V_i$ is the given dry deposition velocity of



corresponding nitrogen speciation. Similar calculation was applied in previously studies (Jung et al., 2013; Luo et al., 2016).

## 3 Results and Discussions

### 3.1 Spatial and temporal variations of water soluble nitrogen species in aerosol

Overall, a significant logarithmic scale seaward decreasing pattern can be seen for all water soluble nitrogen species and WSTN as well in both 2014 and 2015 (Fig. 2). The seaward gradient was mainly caused by continent emission with sea fog influence (Luo et al., 2016), thus, high concentrations in the ECSs (orange in Fig. 2) and low values offshore in background aerosol in the NWPO (blue in Fig. 2). The dust aerosol (pink in Fig. 2) appeared sporadically in the NWPO with higher $NH_4^+$ and $NO_3^-$ (but not WSON) values in general (Table 1 and Fig. 2).

Aerosol WSTN in our observations varied from 21 to 2411 nmol m$^{-3}$ (Table 1 and Fig. 2a and 2b), lower than those in PM$_{10}$ (ranged from 786 to 3000 nmol m$^{-3}$) sampled during spring in the Xi`an city, China (Wang et al., 2013), but higher than those in aerosol sampled in the Sapporo city, Japan (ranged from 20.9 to 108.6 nmol m$^{-3}$; Pavuluri et al., 2015), in the Okinawa Island (ranged from 5 to 216 nmol m$^{-3}$; Kunwar and Kawamura, 2014) and in the North Pacific (ranged from 1.4 to 64.3 nmol m$^{-3}$ in May-July; Hoque et al., 2015). Such wide range of aerosol WSTN illustrated the influence of the distance between sampling locations and emission sources (Matsumoto et al., 2014), seasonality (Kunwar and Kawamura, 2014) and even meteorological conditions, such as sea fog (Luo et al., 2016).

Concentration of marine aerosol WSTN in the ECSs ranged from 444 nmol m$^{-3}$ to 2411 nmol m$^{-3}$ (with volume-weighted mean of 1136 nmol m$^{-3}$) and from 92.9 nmol m$^{-3}$ to 1195 nmol m$^{-3}$ (with volume-weighted mean of 287 nmol m$^{-3}$) in 2014 and 2015 respectively, which were obviously higher than those in dust aerosol (with volume-weighted mean of 242 nmol m$^{-3}$ in 2014 and 154 nmol m$^{-3}$ in 2015) and background aerosol (with volume-weighted mean of 85.6 nmol m$^{-3}$ in 2014 and 42.3 nmol m$^{-3}$ in 2015) collected in the NWPO (Table 1). The air masses backward trajectories (Fig. S4 for 2015 and for 2014 in Luo et al., 2016) revealed the higher aerosol water soluble nitrogen species in the ECSs was from the anthropogenic Nr emission from the east of China (Gu et al., 2012). In addition, frequent formation of sea fog in the ECSs in spring (Zhang et al., 2009) also may enrich the water soluble nitrogen species in sea-fog-modified aerosol by the chemical process during its formation (Luo et al., 2016).

In the NWPO, higher WSTN values in dust aerosol relative to that in background aerosol were observed in the NWPO both in 2014 and 2015, and the WSTN in dust aerosol mainly consisted of $NH_4^+$ and $NO_3^-$ rather than WSON (Table 1, pink circles in Fig. 2), implied that dust can carry more $NH_4^+$ and $NO_3^-$ during long-range transport from the East Asia to the NWPO along with the Asian winter monsoon in spring. The air mass background trajectories of those dust aerosols mainly sourced from the high Nr regions, evidence by dust plume captured by the Lidar browse images from the NASA (Fig. S3 for 2015 and for 2014 in Luo et al., 2016).

The spatial patterns revealed higher concentrations of $NH_4^+$ and $NO_3^-$ in all the aerosols for 2014 than 2015 (Fig. 3a and 3b). The difference between the two years was caused by stronger Asian winter monsoon in 2014 relative to 2015.



Additionally, the cruise in 2014 (17[th] March to 22[th] April) experienced the period of intensive coal/ fossil fuel combustion for heat supply in northern China, yet, the 2015 cruise started on 30[th] March and finished on 3[th]May during recessing period of heat supply. The influence of heat supply can be seen by the atmospheric aerosol optical depth over the heat-supplying areas of China, Xiao et al. (2015) reported that aerosol optical depth is five times higher in heat supplying period than non-supplying period.

Compared with previous study over the same regions in summer (black boxes in Fig. 3a and 3b), our spring data showed higher $NH_4^+$ and $NO_3^-$ in background aerosol (blue boxes in Fig. 3a and 3b). Such distinctively high concentrations observed in spring NWPO suggested a far-reaching influence of anthropogenic emission during monsoon transition. That is, the background aerosol we categorized was influenced by anthropogenic Nr and still not pristine. Apparently, monsoon exerts an important control on annual and seasonal differences in marine aerosol Nr via atmospheric diffusion.

Differing from $NH_4^+$ and $NO_3^-$, concentrations of WSON in background aerosol sampled in the NWPO (blue box in Fig. 3c) in 2014 (average of 10.7±7.0 nmol m$^{-3}$) was similar to that in 2015 (average of 12.7±8.7 nmol m$^{-3}$). Likely the source of WSON in background aerosol did not share the same source with $NH_4^+$ and $NO_3^-$. In the open ocean, apart from the terrestrial aerosol WSON long range transport (Mace et al., 2003; Lesworth et al., 2010), DON in SSW is the most likely source of marine aerosol WSON. For instances, Altieri et al. (2016) reported a strong positive relationship around Bermuda between aerosol WSON and corresponding surface ocean primary productivity. Study in the South Atlantic Ocean also showed 9 times higher WSON in marine aerosol accompanying with higher chlorophyll-*a* in SSW (Violaki et al., 2015). In our observations, higher WSON in background aerosol in spring (blue bar in Fig. 3c) than that in summer (black bar in Fig. 3c) was consistent with higher chlorophyll-*a* concentration in SSW over the NWPO in spring relative to summer (Fig. S5). Such consistency suggested that at least a portion of the aerosol WSON was sourced from the DON in SSW.

## 3.2 Characteristics of CHON molecular compounds

WSON consists of a large number of nitrogen containing compounds and their molecular characteristics can be taken as a tool to identify the potential sources of atmospheric WSON (Altieri et al., 2009, 2012; Wozniak et al., 2014). The CHON molecular formulas were reported to be the most abundant elemental formula compare to others, such as CHONS (composed of C, H, O, N and S) and CHONP (composed of C, H, O, N and P), in both marine aerosol and rainwater (Altieri et al., 2009; Wozniak et al., 2014). CHON formulas with O/C ratio of 0.15 to 0.45, H/C ratio of 1.5 to 2.0 were reported to be peptide-like compounds and functionalized amino acids, which are likely derived from sea spray (Wozniak et al., 2014). Altieri et al (2012) also reported that CHON molecular formulas with characteristics resembles amino acid had the average ratios of O/C and H/C of 0.34 and 1.6, respectively, in seven marine rainwaters.

According to the previous reports (Altieri et al., 2009; Wozniak et al., 2014), we define CHON molecular formulas with O/C of 0.15~0.45 and H/C of 1.5~2.0 as indicators of marine biological sourced compounds. Our results show that the numbers of those biological sourced CHON compounds account for 13 ±3%, 3 ±2% and 19 ±12% of the total numbers of CHON formulas in marine aerosol in the ECSs, dust aerosol and background aerosol in the NWPO in 2015, respectively (Fig.

4). The highest occupation of biological sourced compounds in background aerosol sampled in the NWPO suggested that biological organic nitrogen in SSW can be a source of background aerosol WSON. Such result was consistent with the results in Atlantic Ocean (Altieri et al., 2016).

The lowest occupation was found in the dust aerosol (3 ± 2%) implying the dust aerosol might come from high altitude not yet been influenced much by marine biogenic organics during the sampling periods. On the other hand, although primary production in the ECSs is the highest among areas during the cruise in 2015 (see Fig. S5b), the fraction of marine biological compounds in ECSs' aerosol was not particularly high (Fig.4), which may be attributed to the much stronger influence of continental WSON onto the marine aerosol.

**3.3 Isotopic composition of nitrogen speciation**

Results of $\delta^{15}$N-WSTN for all the aerosols ranged from -10.7 to +5.6‰ (Table 2, Fig 5a and 5b), which consistent with reported range of $\delta^{15}$N-WSTN in precipitation, e.g., -4.2 ~ +12.3‰ in the Baltic Sea (Rolff et al., 2008); -8 ~ +8‰ in the Bermuda (Knapp et al., 2010); -4.9 ~ +3.2‰ in a forest in southern China (Koba et al., 2012); -12.1 ~ +2.9‰ in the Cheju, Korea (Lee et al., 2012). By contrast, our results were lower than the $\delta^{15}$N-WSTN in aerosol sampled in the Sapporo, Japan (+12.2 ~ +39.1‰; Pavuluri et al., 2015) and in the Sapporo forest (+9.0 ~ +26.0‰; Miyazaki et al., 2014). These authors attributed higher isotopic values to the biogenic sources, nitrogenous aerosol aging and fossil fuel combustion. However, WSTN composes of various nitrogen species, the relative proportions of $NH_4^+$, $NO_3^-$ and WSON to WSTN coupled with their isotopic compositions (i.e., $\delta^{15}$N-$NH_4^+$, $\delta^{15}$N-$NO_3^-$ and $\delta^{15}$N-WSON) jointly mediate the variations of aerosol $\delta^{15}$N-WSTN.

Take our case as an example, the $\delta^{15}$N-WSTN in 2014 (-10.7 to +1.0‰) were lower than those in 2015 (–5.6 to +5.6‰) (Table 2, Fig. 6a, 6b and 6c), whereas the ratios of $NH_4^+$/WSTN in 2014 were higher than those in 2015 (Table 1) for all the aerosols. The negative linear relationships between $NH_4^+$/WSTN and $\delta^{15}$N-WSTN for all the aerosols (Fig. 6a, 6b and 6c) may be attributed to higher occupations of $NH_4^+$ in WSTN and low values of $\delta^{15}$N-$NH_4^+$, which was sourced from the anthropogenic and marine emission contains (Altieri et al., 2014; Jickells, 2003; Koba et al., 2012; Liu et al., 2014; Xiao et al., 2012; Yeatman et al., 2001). In fact, low $\delta^{15}$N-$NH_4^+$ values were reported in precipitation collected from many places, such as the Beijing City (ranged from -33.0 to +14.0‰ with an arithmetic mean of -10.8‰; Liu et al., 2014), the Guiyang City in southwestern China (ranged from -38.0 to +5.0‰ with an average of -15.9‰; Xiao et al., 2012), the Gwangju city, south Korea (ranged from -15.9 to +2.9‰ with volume-weighted mean of –6.0‰ in 2007 and -6.8‰ in 2008; Lee et al., 2012) and even in the forest in southern China (range from -18.0 to +0.0‰ with concentration-weighted mean of -7.7‰; Koba et al., 2012). Besides anthropogenic source, atmospheric $\delta^{15}$N-$NH_4^+$ associated with marine air mass was also reported to be low (-8 ~ -5‰, Jickells et al. 2003; -9±8‰,Yeatman et al., 2001; -4.1±2.6‰, Altieri et al., 2014). Summarized together, the averagely low atmospheric $\delta^{15}$N-$NH_4^+$ (-15.9 ~ -4.1‰) is supportive of our findings of higher $NH_4^+$/WSTN and lower $\delta^{15}$N-WSTN in aerosol.

The ratios of $NO_3^-$/WSTN (from 0.27±0.09 to 0.48±0.07) were variable for all the aerosols (Table 1), however, there



were no specifically linear relationships between $NO_3^-$/WSTN ratios and $\delta^{15}$N-WSTN for all aerosols (Fig. 6d, 6e and 6f).
Apparently, $\delta^{15}$N-WSTN was not controlled by the fractional contribution of $\delta^{15}$N-$NO_3^-$. Aerosol $\delta^{15}$N-$NO_3^-$ over the ECSs and the NWPO in 2014 and 2015 ranged from -9.2 to +10.2‰ (Table 2 and Fig. 5c and 5d). All the observed values of $\delta^{15}$N-$NO_3^-$ fall within the previously reported atmospheric $\delta^{15}$N-$NO_3^-$ ranges in land (Elliott et al., 2009; Fang et al., 2011; Felix and Elliott, 2014) and in marine boundary layer (Altieri et al., 2013; Gobel et al., 2013; Hastings et al., 2003; Morin et al., 2009; Savarino et al., 2013). The mass-weighted mean aerosol $\delta^{15}$N-$NO_3^-$ in 2014 (+1.6‰ in the ECSs, -1.1‰ for dust aerosol and -2.6‰ for background aerosol sampled in the NWPO) were similar to those in 2015 (+1.9‰ in the ECSs, -3.8‰ for dust aerosol and -1.1‰ for background aerosol sampled in the NWPO) (Table 2) for all the aerosols, suggested that aerosol $NO_3^-$ in both 2014 and 2015 experience similar origins and atmospheric chemical pathways.

However, there were positive linear relationships between ratios of WSON/WSTN and $\delta^{15}$N-WSTN for all the aerosols can be seen (Fig. 6g, 6h and 6i). It implied that aerosol $\delta^{15}$N-WSON may have positive $\delta^{15}$N values. For the WSON in marine aerosol, it either originated from terrestrial long-range transport or sourced from the DON in SSW as discussed in Section 3.1 and Section 3.2. Terrestrial aerosol $\delta^{15}$N-WSON was reported in a wide range (-15.0 ~ +14.7‰) with mean values from -3.7 to +5.0‰, and $\delta^{15}$N-WSON in marine aerosol with more positive $\delta^{15}$N (Fig. S1). In addition, our own observations for $\delta^{15}$N-DON in SSW (5m depth) showed positive $\delta^{15}$N in the ECSs (varied from +5.1 to +12.9‰, with an average +7.9±2.3‰) and the NWPO (ranged from +1.9 to +11.6‰, with an average +5.7±2.0‰) (Fig. 7a). At the same time, concentrations of DON in our observations ranged from 4.4 to 11.8 μmol L$^{-1}$ (Fig. 7b), which within the reported concentration of DON in global SSW (Knapp et al., 2011; Letscher et al., 2013; Li et al., 2009; Lønborg et al., 2015; Van Engeland et al., 2010). Such a high concentration of DON in SSW may be ejected into the atmosphere during the bubble bursting (Wilson et al., 2015).

To better clarify the sources of aerosol WSON, we removed the aerosol $NO_3^-$ and its $\delta^{15}$N effect from the WSTN and its $\delta^{15}$N-WSTN, respectively, by mass balance. The remained $NH_4^+$ and WSON was defined as reduced nitrogen (RN = $NH_4^+$ + WSON). The $\delta^{15}$N-RN ranged from -11.8 to +7.4‰ for all aerosols in both 2014 and 2015 (Table 2), which consistent with reported $\delta^{15}$N-RN in precipitation (-12.6 ~ +7.8‰) collected in Bermuda, Atlantic (Knapp et al., 2010). Three important end-members, compiled atmospheric $\delta^{15}$N-$NH_4^+$ (-15.9 ~ -4.1‰, green bars) and the continental $\delta^{15}$N-WSON (-3.7 ~ +5.0‰, gray bars) versus $\delta^{15}$N-DON observed for SSW in our cruise (+7.9±2.3‰ in the ECSs and +5.7±2.0‰ in the NWPO, red dots with error bars) were together added into Figure 8 for discussion. Note that nearly all the aerosol $\delta^{15}$N-RN in 2014 was lower than those in 2015, which may attribute to higher $NH_4^+$ concentration in 2014 than in 2015 (Table 1). Moreover, the higher ratios of WSON/RN, the higher values of $\delta^{15}$N-RN for all the aerosols can be seen in Figure 8. The high values of $\delta^{15}$N between continental WSON and marine SSW DON are the possible causes for such positive patterns. Thus, although higher WSON/RN values accompany with higher $\delta^{15}$N-RN, we may still not conclude a significant SSW DON contribution to aerosol WSON, such as the case for the ECSs with intensive continental influence and the case for dust aerosols since FT-ICRMS results gave less numbers of marine sourced organics. In the open ocean, to some extent, background aerosol WSON was more likely influenced by SSW DON judging by nitrogen isotopic information and CHON



molecular formula which mentioned in section 3.2.

Note that some data points collected in 2015 for open ocean case in Figure 8b and 8c fell outside the mixing field
deviating toward higher $\delta^{15}$N-RN values. The outside values may be attributable to aerosol WSON aging process, in-cloud
scavenging (Altieri et al., 2016), complex atmospheric chemistry reactions (i.e., photolysis of organic nitrogen into
ammonium; Paulot et al., 2015) and the formation of second organic aerosol by the ammonia and nitrate with volatile
organic compounds (De Haan et al., 2010; Fischer et al., 2014) during the long-range transport, of which $\delta^{15}$N fractionation
and enrichment of $^{15}$N might occur. More studies are needed to explore nitrogen transformation processes from isotopic
perspective.

### 3.4 Dry Nr deposition and its biogeochemical role

The dry depositions of aerosol $NH_4^+$, $NO_3^-$ and WSON were summarized in Table 3. The calculated depositional fluxes
of water soluble nitrogen species in the ECSs were significant higher than those in NWPO (Fig. 9). Averaged dry depositions
of $NH_4^+$ and $NO_3^-$ in 2014 were 2 ~ 5 times higher than those of 2015 (Table 3) for all the aerosols. The dry deposition
fluxes of $NH_4^+$ and $NO_3^-$ in dust aerosol were obviously higher than those in background aerosol in the NWPO (Table 3,
Fig. 9a, 9b, 9c and 9d). The comparison of dry fluxes with other similar studies have been discussed in Luo et al. (2016)
specifically for 2014, here we focus on the influence of atmospheric Nr deposition on the nitrogen cycle of upper ocean.

The mass-weighted means of $\delta^{15}$N-WSTN for 2014 and 2015 were -5.5‰ and +0.9‰ in the NWPO, respectively,
(Table 2), similar to the $\delta^{15}$N of $N_2$-fixation (~-2 ~ 0‰; summed by Knapp et al., 2010). Thus, both the atmospheric Nr
deposition and $N_2$-fixation to the surface ocean can low down the $\delta^{15}$N-$NO_3^-$ in the subsurface after re-mineralization. Such
phenomenon had been reported in the thermocline of the Atlantic (Knapp et al., 2010) and the South China Sea (Yang et al.,
2014). Here, we follow the previous approach in both Atlantic (Knapp et al., 2010) and South China Sea (Yang et al., 2014)
to estimate the proportion(X) of lowing the thermocline $\delta^{15}$N-$NO_3^-$ in the NWPO which may be caused by the atmospheric
dry deposition of Nr:

$$X = (F_{AD} \times \Delta\delta N_{BT\text{-}AD}) / (F_{NF} \times \Delta\delta N_{BT\text{-}NF} + F_{AD} \times \Delta\delta N_{BT\text{-}AD})$$

where the $F_{AD}$ are the fluxes of atmospheric dry deposition (the total Nr fluxes: $NH_4^+$+ $NO_3^-$ + WSON) for the background
aerosol in the NWPO were 55.3 and 36 µmol N m$^{-2}$ d$^{-1}$ for 2014 and 2015, respectively), and the $F_{NF}$ are nitrogen fixation in
the subtropical North Pacific Ocean (84 ~ 140 µmol N m$^{-2}$ d$^{-1}$; Karl et al., 1997). The $\Delta\delta N_{BT\text{-}AD}$ is the difference between the
$\delta^{15}$N-$NO_3^-$ below the thermocline in the NWPO (~+5.6‰, Kao, unpublished data) and the mass-weighted mean
$\delta^{15}$N-WSTN in background aerosol (-5.5 ~ +0.8‰); the $\Delta\delta N_{BT\text{-}NF}$ represents the discrepancy between the $\delta^{15}$N-$NO_3^-$ below
the thermocline in the NWPO (+5.6‰) and the $\delta^{15}$N of $N_2$-fixation (~-2 ~ 0‰). The calculation implies that the
atmospheric Nr dry deposition flux can account for 14% ~ 58% of low the $\delta^{15}$N-$NO_3^-$ in the NWPO thermocline during the
spring. If the dust events (total fluxes during the dust plume with mass-weighted mean sampled in the NWPO were 187 and
136 µmol N m$^{-2}$ d$^{-1}$ for 2014 and 2015) taken into account, the depression of atmospheric dry deposition to the $\delta^{15}$N-$NO_3^-$ in



the NWPO may be more pronounced.

However, there were some uncertainties about the calculation. First, the wet deposition flux of WSTN and the ratio of wet deposition to the dry deposition are unknown during our study periods in the NWPO. A previous study in the subarctic western North Pacific Ocean during the summer periods reported that the ratio of total dry deposition flux of aerosol $NH_4^+$ and $NO_3^-$ to wet deposition is 0.36 (Jung et al., 2013). However, our observation carried out during the Asian winter

monsoon when the dry deposition flux of $NH_4^+$ and $NO_3^-$ (24.4 ~ 47.8 µmol N m$^{-2}$ d$^{-1}$; Table 3) were significant higher than those in summer (4.9 µmol N m$^{-2}$ d$^{-1}$; Jung et al. 2013), and the precipitation in spring is also lower than that in summer (Dorman and Bourke, 1979). Thus, it is hard to estimate the total deposition during the spring over the NWPO. Second, we using the $N_2$-fixation fluxes in station ALOHA substitute for our study regions. The previous studies reported that $N_2$-fixation rate in ocean influenced by the concentrations of iron, phosphorus, and temperature (Mulholland and Bernhardt,

2005; Needoba et al., 2007), and the differences of global measured $N_2$-fixation rate as high as two order of magnitude (Montoya et al., 2004). The third, as discussed in Section 3.1 - 3.3, parts of WSON in background aerosol originate from the DON in SSW, the total ($NH_4^+$ + $NO_3^-$ + WSON) dry deposition fluxes to the NWPO may overestimate.

## 4 Conclusions

Concentrations of water-soluble total nitrogen (WSTN), nitrate ($NO_3^-$) and ammonium ($NH_4^+$) as well as stable nitrogen

isotopes of $\delta^{15}$N-WSTN and $\delta^{15}$N-$NO_3^-$ were measured in marine aerosol sampled from the ECSs to the NWPO in spring of 2014 and 2015. Dissolve organic nitrogen (DON) and $\delta^{15}$N-DON also were analyzed in SSW (5m depth) collected along with the cruise in spring of 2015. The highest concentrations of water soluble nitrogen species in aerosol sampled in the ECSs suggest significant influence of anthropogenic emission on aerosol Nr. Higher $NO_3^-$ and $NH_4^+$ in 2014 for all aerosols than those in 2015 may be attributed to the stronger Asian winter monsoon in 2014 as well as the intensity of heat

supply in spring in North of China. However, concentration of WSON in background aerosol collected in the NWPO was comparable in 2014 and 2015. Together with satellite chlorophyll-*a* concentration in SSW and CHON molecular formulas in aerosol, we infer that WSON in background aerosol may come from DON in SSW.

There were negative linear relationships between ratios of $NH_4^+$/WSTN and $\delta^{15}$N-WSTN for all aerosol. On the contrary, positive linear relationships between ratios of WSON/WSTN and $\delta^{15}$N-WSTN were observed. The distinctive

occupations of nitrogen species and isotopic compositions suggest that aerosol $\delta^{15}$N-WSTN values were mediated synergistically by $NH_4^+$ and WSON in our observations. Meanwhile, our isotope mixing model indicated that DON in SSW is likely to be one of the sources of aerosol WSON, especially in the open ocean. There are many unknowns about the Nr in marine boundary layer and surface sea water, let alone to the Nr in air-sea interface. Studies about the air-sea Nr exchange are needed in future.


*Acknowledgements.* This research was funded by the National Natural Science Foundation of China (NSFC U1305233), the



Major State Basic Research Development Program of China (973 program) (NO.2014CB953702 and 2015CB954003), the National Natural Science Foundation Committee and Hong Kong Research Grants Council Jointly Foundation (NSFC-RGC 41561164019), the National Natural Science Foundation of China (NSFC 41763001), Doctoral Scientific Research
Foundation of East China University of Technology (NO. DHBK2016105), Science and technology project of Jiangxi Provincial Department of Education (NO. GJJ160580) and Scientific Research Foundation of East China University of Technology for Science and Technology Innovation Team (NO. DHKT2015101). Dr. Thorsten Dittmar and Dr. Jutta Niggemann from the Carl von Ossietzky University of Oldenburg are acknowledged for their help in the FT-ICR-MS measurement. This is MEL (State Key Laboratory of Marine Environmental Science) publication 2017209.

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





**Table 1: Ranges and means of concentrations of WSTN, $NH_4^+$, $NO_3^-$, WSON and RN in aerosol.**

| | | WSTN (nmol m⁻³) | | | $NH_4^+$ (nmol m⁻³) | | | $NH_4^+$/ WSTN | $NO_3^-$ (nmol m⁻³) | | | $NO_3^-$/ WSTN | WSON (nmol m⁻³) | | | WSON/ WSTN | RN (nmol m⁻³) | | |
|---|---|---|---|---|---|---|---|---|---|---|---|---|---|---|---|---|---|---|---|
| | | Range | Mean[a] | Mean[b] | Range | Mean[a] | Mean[b] | | Range | Mean[a] | Mean[b] | | Range | Mean[a] | Mean[b] | | Range | Mean[a] | Mean[b] |
| ECSs (Sea fog) | 2014 | 444 ~ 2411 | 1126 | 1136 | 228 ~ 777 | 442 | 437 | 0.42±0.09 | 160 ~ 1118 | 536 | 550 | 0.48±0.07 | 23 ~ 517 | 149 | 148 | 0.10±0.06 | 267 ~ 1294 | 591 | 585 |
| | 2015 | 92.9 ~ 1195 | 321 | 287 | 25.7 ~ 564 | 126 | 113 | 0.35±0.12 | 30.1 ~ 239 | 93.2 | 87.6 | 0.35±0.15 | 6.9 ~ 392 | 102 | 85.7 | 0.29±0.18 | 32.6 ~ 956 | 228 | 199 |
| NWPO (Dust) | 2014 | 205 ~ 297 | 245 | 242 | 94.4 ~ 163 | 138 | 137 | 0.56±0.07 | 78.6 ~ 145 | 100 | 98.3 | 0.41±0.05 | 7.7 ~ 16.9 | 11.2 | 11 | 0.05±0.02 | 111 ~ 173 | 145 | 144 |
| | 2015 | 81.4 ~ 340 | 147 | 154 | 20.7 ~ 143 | 58.3 | 61.4 | 0.39±0.13 | 34.7 ~ 126 | 57.2 | 58.6 | 0.41±0.1 | 5.8 ~ 80.7 | 31.7 | 34.4 | 0.21±0.13 | 45.0 ~ 214 | 89.9 | 95.8 |
| NWPO (Bgd.) | 2014 | 31.4 ~ 411 | 88.8 | 85.6 | 16.1 ~ 244 | 54 | 51.8 | 0.60±0.11 | 6.4 ~ 166 | 25.8 | 25.1 | 0.27±0.09 | 0.9 ~ 27.1 | 10.7 | 9.7 | 0.14±0.08 | 19.2 ~ 245 | 63.1 | 60.5 |
| | 2015 | 21.0 ~ 68.7 | 41.7 | 42.3 | 6.9 ~ 29.5 | 16.3 | 15.5 | 0.41±0.15 | 2.8 ~ 35.1 | 12.7 | 13.4 | 0.29±0.13 | 1.0 ~ 40.3 | 12.7 | 13.4 | 0.30±0.16 | 15.2 ~ 48.2 | 29 | 28.9 |

a, arithmetic mean
b, volume-weighted mean



**Table 2: Ranges and means of stable nitrogen isotopes of WSTN, $NO_3^-$ and RN in aerosol.**

| | | $\delta^{15}N\text{-WSTN}$ | | | $\delta^{15}N\text{-}NO_3^-$ | | | $\delta^{15}N\text{-RN}$ | | |
| --- | --- | --- | --- | --- | --- | --- | --- | --- | --- | --- |
| | | Range | Mean[a] | Mean[b] | Range | Mean[a] | Mean[b] | Range | Mean[a] | Mean[b] |
| ECSs (Sea fog) | 2014 | -5.3 ~ -1.7 | -3.4 | -3.4 | -1.7 ~ +4.3 | +1.0 | +1.6 | -11.8 ~ -2.5 | -7.3 | -7.9 |
| | 2015 | -4.3 ~ +0.9 | -1.1 | -2.1 | -1.3 ~ +10.2 | +2.8 | +1.9 | -5.2 ~ -0.6 | -3.3 | -3.7 |
| NWPO (Dust) | 2014 | -6.9 ~ -3.1 | -4.7 | -4.4 | -3.0 ~ +1.3 | -1.0 | -1.1 | -9.4 ~ -5.8 | -7.1 | -7.3 |
| | 2015 | -3.0 ~ +0.8 | -1.5 | -1.3 | -7.3 ~ -2.1 | -3.9 | -3.8 | -2.7 ~ +5.5 | +0.6 | +0.3 |
| NWPO (Bgd.) | 2014 | -10.7 ~ +1.0 | -5.6 | -5.5 | -7.6 ~ +4.3 | -2.3 | -2.6 | -11.7 ~ +1.5 | -6.9 | -6.7 |
| | 2015 | -5.6 ~ +5.6 | +0.8 | +0.9 | -9.2 ~ +1.2 | -1.6 | -1.1 | -5.0 ~ +7.4 | +1.9 | +1.7 |

a, arithmetic mean
b, mass-weighted mean



**Table 3: The dry deposition fluxes of water soluble nitrogen species.**

| | | NH$_4^+$ $\mu$mol N m$^{-2}$ d$^{-1}$ | | | NO$_3^-$ $\mu$mol N m$^{-2}$ d$^{-1}$ | | | WSON $\mu$mol N m$^{-2}$ d$^{-1}$ | | |
|---|---|---|---|---|---|---|---|---|---|---|
| | | Range | Mean[a] | Mean[b] | Range | Mean[a] | Mean[b] | Range | Mean[a] | Mean[b] |
| ECSs(Sea fog) | 2014 | 19.7 ~ 67.2 | 38.1 | 37.8 | 277 ~ 1931 | 926 | 951 | 19.7 ~ 446 | 127 | 128 |
| | 2015 | 2.2 ~ 48.7 | 14.6 | 13.9 | 52.1 ~ 412 | 176 | 169 | 5.9 ~ 339 | 104 | 91.8 |
| NWPO ( Dust) | 2014 | 8.2 ~ 14.1 | 11.9 | 11.9 | 136 ~ 250 | 172 | 170 | ~ 14.6 | 6.5 | 5.5 |
| | 2015 | 1.8 ~ 12.4 | 5.0 | 5.3 | 60.0 ~ 218 | 98.8 | 101 | 5.0 ~ 69.7 | 27.3 | 29.7 |
| NWPO (Bgd.) | 2014 | 1.4 ~ 21.1 | 4.7 | 4.5 | 11.0 ~ 287 | 44.6 | 43.3 | 0.8 ~ 23.4 | 7.6 | 7.5 |
| | 2015 | 0.6 ~ 2.6 | 1.4 | 1.3 | 4.9 ~ 60.6 | 21.9 | 23.1 | 0.9 ~ 34.8 | 10.9 | 11.6 |

a, arithmetic mean
b, volume-weighted mean





**Figure 1: Regional wind streamlines (in unit m s⁻¹) at 1000 hPa in the period of the Asian winter monsoon (a, March and April in 2014 and b, March and April in 2015) based on the NCEP data set, and cruise tracks also as shown (orange, pink and blue indicate sea fog, dust and background days during the cruise). Aerosols number and the collection ranges are shown with orange, pink and blue numbers and lines. The black open circles in the (b) indicate the locations of surface sea water (5m) during the cruise of 2015.**



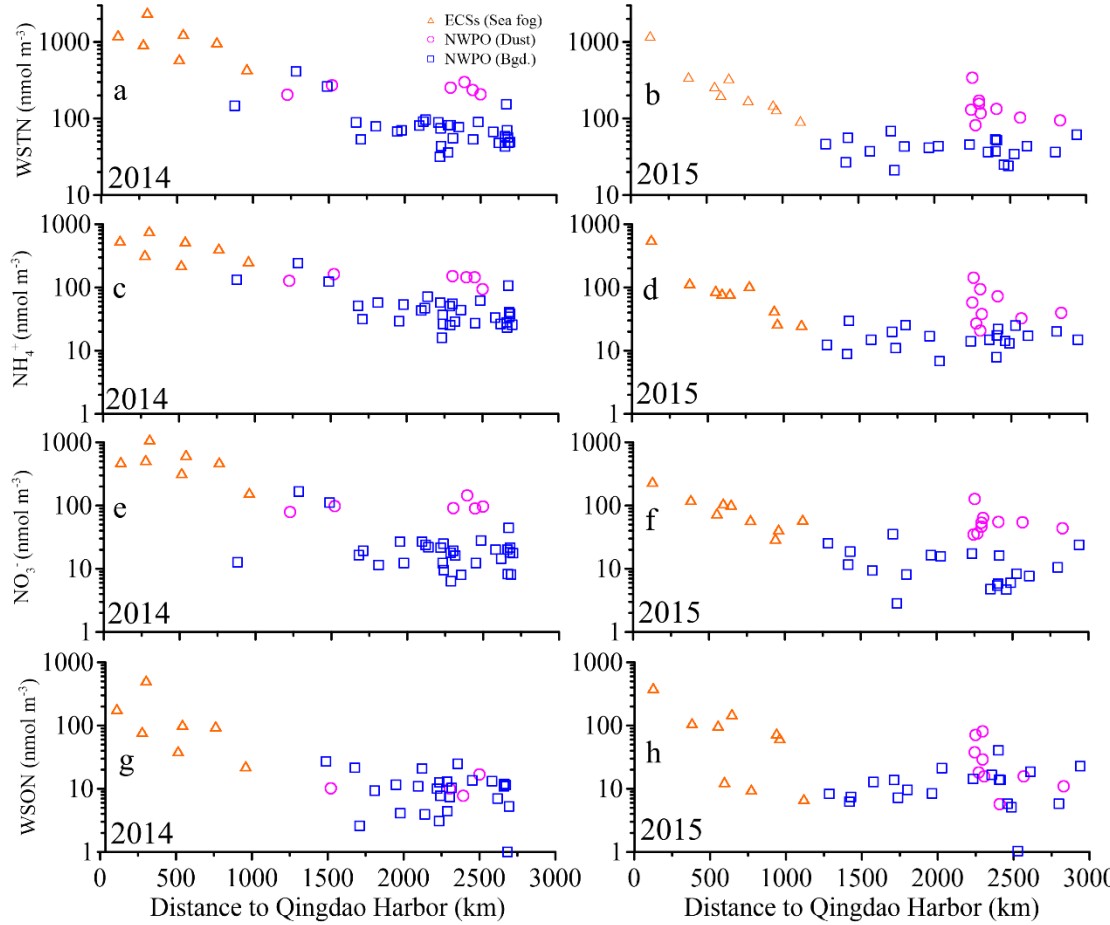

**Figure 2: Concentrations of aerosol WSTN (a and b), $NH_4^+$ (c and d), $NO_3^-$ (e and f) and WSON (g and h) in the ECSs (orange open triangles) and in the NWPO (pink open circles for dust aerosol and blue open squares for background aerosol) distance to Qingdao Harbor in cruise of 2014 and 2015, respectively.**





**Figure 3: Box plots for spring concentrations of aerosol $NO_3^-$(a), $NH_4^+$(b) and WSON(c) in the ECSs (orange boxes) and in the NWPO (pink boxes for dust aerosol and blue boxes for background aerosol), and summer aerosol (black boxes, Miyazaki et al., 2011). The large boxes represented the interquartile range from the 25th to 75th percentile. The line inside the box indicates the median value. The whiskers extend upward to the 90th and downward to the 10th percentile.**



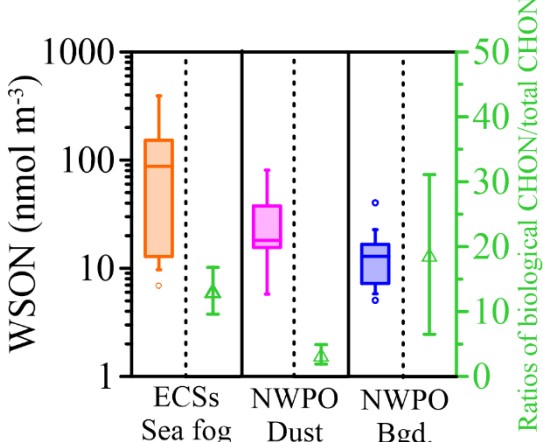

**Figure 4: Box plots for concentrations of WSON and the mean ratios of defined biological CHON to total CHON in 2015 aerosol.**

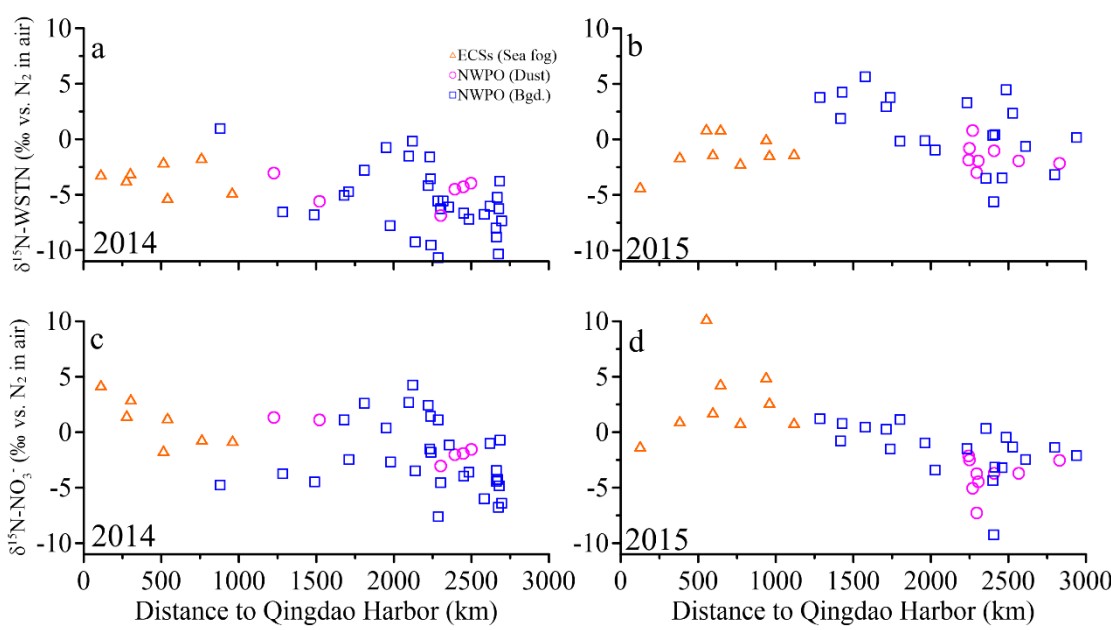

**Figure 5: Aerosol δ¹⁵N-WSTN (a and b) and δ¹⁵N-NO₃⁻ (c and d) in the ECSs (orange open triangles), and in the NWPO (pink open**
10 **circles for dust aerosol and blue open squares for background aerosol) distance to Qingdao Harbor in cruise of 2014 and 2015,**
**respectively.**




**Figure 6:** Scatter plots of $\delta^{15}$N-WSTN against corresponding to the ratios of $NH_4^+$/WSTN in aerosol sampled in the ECSs (a), dust aerosol (b) and background aerosol (c) collected in the NWPO; aerosol $\delta^{15}$N-WSTN against corresponding to the ratios of $NO_3^-$/WSTN in the ECSs (d), dust aerosol (e) and background aerosol (f) in the NWPO; and aerosol $\delta^{15}$N-WSTN against corresponding to the ratios of WSON/WSTN in the ECSs (g), dust aerosol (h) and background aerosol (i) in the NWPO. The solid and open symbols indicate the aerosol sampled in 2014 and 2015, respectively.



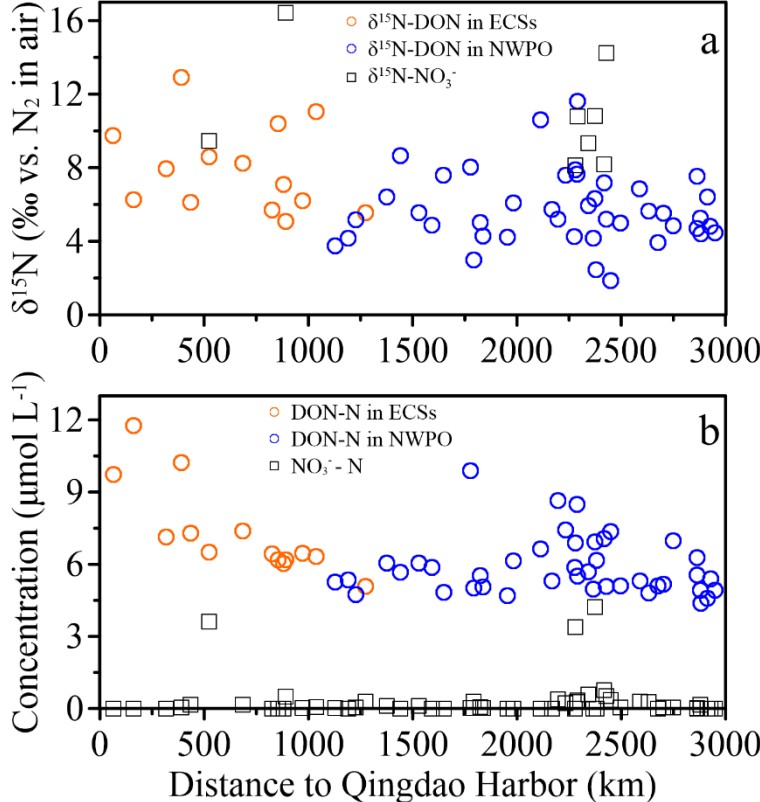

**Figure 7: (a) Concentrations of DON (open circles) and $NO_3^-$ (open squares), (b) $\delta^{15}N$-DON (open circles) and $\delta^{15}N$-$NO_3^-$ (open squares) in SSW distance to Qingdao Harbor in cruise of 2015.**





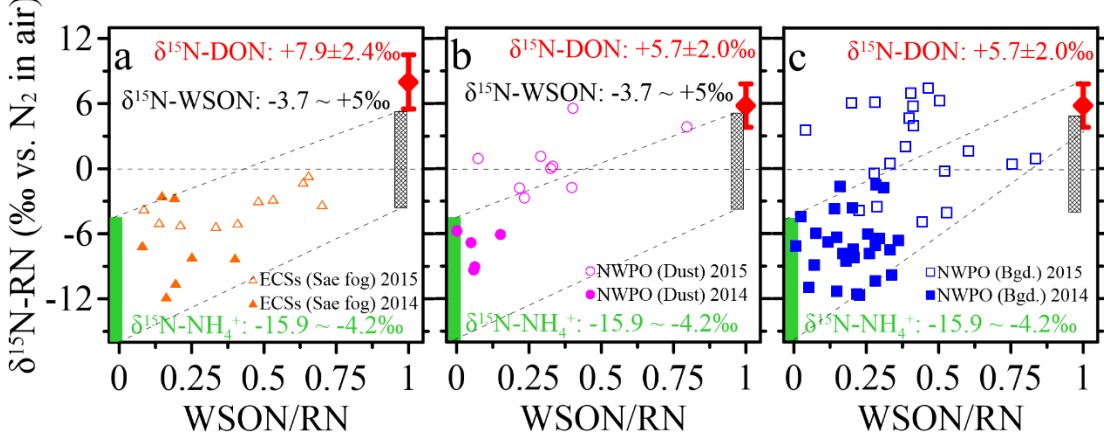

**Figure 8: Scatter plots of aerosol δ15N-RN against corresponding to the ratios of WSON/RN in the ECSs (a) and in the NWPO (dust aerosol (b) and background aerosol (c)). The green bar indicates the sources of anthropogenic, terrestrial and oceanic δ15N-NH₄⁺, the gray bar indicates the sources of terrestrial and anthropogenic δ15N-WSON and the red bar indicates the δ15N-DON in SSW.**





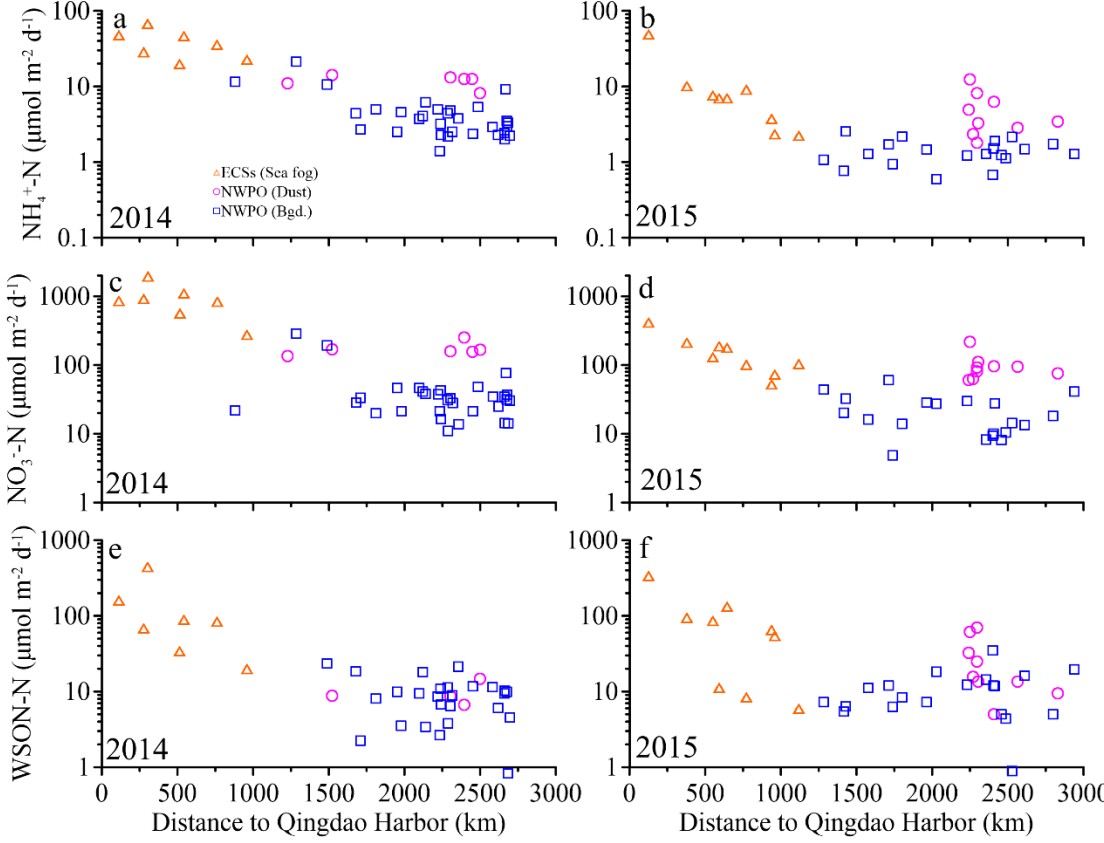

**Figure 9: Dry deposition of aerosol $NH_4^+$-N (a and b), $NO_3^-$-N (c and d) and WSON-N (e and f) in the ECSs (orange open triangles) and in the NWPO (pink open circles for dust aerosol and blue open squares for background aerosol) distance to Qingdao Harbor in cruise of 2014 and 2015, respectively.**

