# Peer review of "Sources of reactive nitrogen in marine aerosol over the Northwest Pacific Ocean in spring"

_Atmospheric Chemistry and Physics, 2017_

## Referee Comment (RC1) · Anonymous Referee #1 · 29 Nov 2017

This paper presents a very interesting and important data set on Nr aerosol concentrations as well as the $\delta$15N of total N and NO3- in aerosols. The aerosols were collected on ship transects from China to the Northwest Pacific Ocean, and as such are presented as marine aerosols. This is a dataset that should be published and it is important work. However, given the incredibly strong anthropogenic source strength of Nr in this region, I do not think these are representative of marine aerosols. Rather, this is a presentation of the impact of continental/anthropogenic aerosols on the coastal/near shore marine environment. It is a subtle, but important distinction. The review is presented below as major and minor comments.

Major Comments: Section 2.2.5: The aerosol extracts do not contain salts, which is typically why SPE is used prior to FT-ICR MS analysis. It is not clear why this procedure was followed and some justification should be provided? It will lead to loss of organic carbon and organic nitrogen, indeed the % recoveries are < 50% and it does not seem appropriate. FT-ICR MS analysis using negative ion mode means it is not comparable to Altieri et al., or Wozniak et al. In those studies they analyzed samples in the positive ion mode. The negative ion mode would detect organonitrate compounds, whereas the amine and amino-acid compounds are detected in the positive ion mode. The listed elemental ratios based on positive ion mode analysis are not at all applicable in this work. Given the use of SPE and the negative ion mode analysis, the FT-ICR MS analysis needs to be removed from the manuscript, or interpreted in a completely different manner. I agree with page 1 line 24-26 – the anthropogenic signal is so strong that there is no way these aerosols represent a background signal at all. The classification needs to be explained in more detail, and changed to something more appropriate. It seems impossible that aerosols classified as "background" could have such high concentrations of Nr. It is more likely that all of these aerosol samples are heavily influenced by anthropogenic pollution, with the signal declining as the polluted air is mixed with clean air off shore. Any local marine signal would be swamped by the large continental sources. The similarity in 2014 and 2015 WSON concentrations is interesting given different chlorophyll fields. It is critical that some basic sampling information such as aerosol size is presented. In addition, the authors need to provide information on field blanks and procedural blanks, especially for WSON and ammonium concentrations. In Section 2.2.3, please explain the recoveries of WSTN and TDN? How does n=6 if there were 44 and 39 aerosol samples analyzed? What dry deposition velocities are used? There are large uncertainties associated with dry deposition velocities, which are size specific. If the aerosols are not size segregated, how can you apply a size-specific dry deposition velocity? These are going to be highly uncertain estimates and should be treated as such. Page 7 line 99-101: It's not clear what size the aerosols are, and so this comparison is difficult to understand. Regardless, these concentrations are incredibly high, and indicative of strong pollution sources. If the authors want to claim that any of these aerosols are "background" aerosols, they

[Figure]

need to find other background sites that have such high Nr concentrations. Also, the data are very difficult to see on the log scale, it's a very large range of concentrations to present on one figure. This should be separated somehow. It's not clear how the dust aerosols were identified as such and some explanation is required. Page 7 line122, you can't tell the difference between 2014 and 2015 in the figures due to the log scale. A different way of presenting the data would help. I agree with the authors conclusion on page 8 paragraph 2 – the "background" aerosols should be re-labeled as it is very misleading. Page 8 paragraph 3 – it is important to note that the surface seawater DON is the most likely source of primary WSON aerosols, but secondary WSON aerosols have many other sources, including e.g., surface ocean VOC emissions that go on to oxidize in the atmosphere and form secondary N-containing SOA. Section 3.2 should be removed or reinterpreted given the focus on negative mode CHON compounds identified here. Page 9 section 3.3 second paragraph. There is a large difference in WSTN $\delta$15N from 2014 to 2015 in these aerosols. This should be discussed. Figure 6 and discussion thereof: This is not a valid approach to understanding what is driving the $\delta$15N of WSTN. A cross-plot of $\delta$15N-NO3- vs $\delta$15N-WSTON would provide more information on the influence of nitrate on the total N isotopic composition. Looking at figure 5, it looks like nitrate $\delta$15N is a main control on the $\delta$15N-WSTN. The lack of correlation between $\delta$15N WSTN and the relative concentration of NO3- is not useful. The relationship between the $\delta$15N RN and the relative concentrations of NH4+ would be useful, but is not presented. Page 10 paragraph line 300, it is also a possibility that the aerosol WSON is secondary organic aerosol, which may have had its $\delta$15N altered by transport or chemical reactions. This is a very over-simplified approach to the interpretation of the $\delta$15N-WSON data. Section 3.4 is too speculative given the limited information presented. Are the ammonium, nitrate, and WSON concentrations statistically different from 2014 to 2015 and between the three classifications? Is there a statistically significant relationship between the $\delta$15N of DON in seawater or $\delta$15N NO3- in seawater and the $\delta$15N of TN, NO3- , or RN in the aerosols? Minor Comments: Table 1. How are aerosol concentrations volume weighted? Is this a mass

weighted average? Figure 1. It is not clear what "regional wind streamlines" are or where they came from. The blue on the background of the figure makes it difficult to see the symbols. Figure 3. The caption says a is nitrate and b is ammonium, but they are labeled in the opposite manner. Figure 4 should be removed. Figure 7. The caption says a is concentration and b is $\delta$15N but the plots are the opposite. Abstract: Line 14 insert "of" between transport and anthropogenic, line 15 "continents may exert a profound impact", line 16 should read "surface ocean" instead of "marine biogenic", line 18 do the authors mean in the open ocean or do they mean in the atmosphere?, line 26 are the concentrations statistically higher in 2014? If so this should be presented in the text. Introduction: Define SSW on first use. Page 3 paragraph 1 should clearly state that they are referring to primary WSON aerosols. Page 8 second paragraph. It's not clear what is meant by "atmospheric diffusion"

---

## Referee Comment (RC2) · Anonymous Referee #2 · 5 Dec 2017

This manuscript describes ship-board measurements of marine aerosols collected during two cruises around the East China Sea and the northwestern Pacific Ocean in 2014 and 2015. In this manuscript, authors reported concentrations of water-soluble total nitrogen (WSTN), water-soluble organic nitrogen (WSON), nitrate (NO3−) and ammonium (NH4+), and values of $\delta$15N-WSTN and $\delta$15N-NO3− in aerosols as well as dissolved organic nitrogen (DON) and NO3− concentrations and $\delta$15N-DON and $\delta$15N-NO3− values in sea surface water, which provide good indications where future studies can understand possible sources of atmospheric WSON and air-sea exchange of N species. I believe that the contents, including data, of the manuscript should be eventually published because of scarcity of atmospheric WSON observation and its significance in biogeochemical N cycle. However, the manuscript is NOT publishable

in its current format. I recommend publication of the manuscript after a major revision and the improvement of English. Overall, authors were successful in addressing their ideas in the Results and discussions section, though. A number of changes are recommended below.

Major points

1. The title of this manuscript is "Sources of reactive nitrogen in marine aerosol over the Northwest Pacific Ocean in spring"; however, the authors mainly described the spatial distributions and concentrations of atmospheric reactive N species and potential sources of WSON. I recommend the authors to describe in their manuscript the sources of atmospheric inorganic N species, although they are relatively well-known compared to that of WSON.

2. (Page 4, line 98−116) It is not clear that how many, what kind of aerosol samplers and filters were used during the cruises, and how avoid ed contamination from ship's exhaust. Was a impactor used to separate PM2.5 and PM10? More detail information on sampling method should be described in the manuscript, although the authors referred to Luo et al. (2016). In general, a pre-combusted glass fiber or a quartz filter is used for determination of WSON. If the authors used the same method for aerosol sampling described in Luo et al. (2016), the authors should explain about the treatment of aerosol filter samples and field blank concentrations and blank correction, because Luo et al. (20016) used a Whatman 41 cellulose filter. For determination of DON in sea surface water, the authors mentioned that a 0.2 $\mu$m filter was used to remove particulate matters in sea surface water. Usually, a pre-combusted GF/F filter is used to remove particulate matter and minimize the influence of organic matter from the filters on DON concentration in seawater. Please update that what kind of filter was used for filtration of seawater sample. I am also wondering if any consensus reference material (CRM, e.g., deep Florida Strait water from Hansell lab, University of Miami) was used during DON measurement to check the accuracy of analysis.

3. (Page 6, line 188−Page 7, line 191) Dry deposition velocity. It is unclear if marine aerosols are segregated into PM2.5 and PM10 during the aerosol sampling as mentioned in question 2. Although size distributions of atmospheric N species can vary on meteorological conditions, it is known that, in the marine atmosphere, both atmospheric NH4+ and WSON primarily exist on fine mode aerosols, whereas atmospheric NO3− is predominantly associated with coarse mode aerosols (e.g., Nakamura et al., 2006). I recommend the authors to describe more detail that what dry deposition velocity was used for each N species.

4. (Page 8, line 228−232) The authors compared their NH4+ and NO3− concentrations with those by Miyazaki et al. (2011) to explain why higher concentrations of inorganic N species were observed during the period of this study (spring). The authors mentioned that the study by Miyazaki et al. (2011) was carried out over the same regions. I doubt about it. The cruise by Miyazaki et al. (2011) was conducted from 44°N to 10°N along 155°E, which covers the subarctic to subtropical northwestern Pacific region. Although the study by Miyazaki et al. (2011) was carried out in summer, different sampling season is not the only reason why the authors observed high inorganic N species during their study period.

5. (Page 8, line 233−242) The authors described that "Likely the source of WSON in background aerosol did not share the same source with NH4+ and NO3−" (line 234−235), as if DON in sea surface water is the only source of atmospheric WSON in the open ocean. What is the grounds for this? Because high atmospheric WSON and inorganic N species concentrations were observed in the East China Sea and inorganic N was also detected in the open ocean, the long range transport of anthropogenic WSON to the open ocean should be considered.

6. (Page 8, line 252−Page 9, line 263) The results of characteristics of CHON molecular compounds shows that 13%, 3% and 19% of marine aerosols collected in the East China Sea, northwestern Pacific Ocean during dust period and northwestern Pacific Ocean during non dust period, were derived from biological sources, respectively.

Does this mean that 87%, 97%, and 81% of marine aerosols collected in the same regions were affected by anthropogenic sources? It seems like the contribution of biogenic sources to atmospheric WSON is still low in the open ocean. What is the contribution of biologically-derived atmospheric WSON in the other oceanic regions?

7. (Page 11, line 331−Page 12, line 355) The authors described that atmospheric reactive N dry deposition flux can account for 14%−58% of the low $\delta$15N-NO3− in the northwestern Pacific Ocean during the spring. It is surprising to me that atmospheric reactive N deposition has a significant influence on $\delta$15N-NO3− values. My question is that dry deposition of atmospheric reactive N is strong enough to affect or change $\delta$15N-NO3− values below the thermocline in the northwestern Pacific Ocean? What is the depth of thermocline in the northwestern Pacific Ocean in the spring season? I recommend the authors to estimate the contribution of atmospheric reactive N dry deposition to primary production in their study area. I think most primary production in the East China Sea and northwestern Pacific Ocean is controlled by nutrients in seawater, which implies that main factor for controlling $\delta$15N-NO3− values in the ocean is marine N cycle.

Minor points

1. (Page 5, line 136−137) How did the authors obtain the recovery efficiency (i.e., 95−105% (n = 6)) of WSTN and TDN?

2. (Page 5, 155−156) The authors mentioned that the extraction efficiency on a carbon basis was on average 46 ± 24% (n = 44). Does it mean that 64% of organic compounds in the extract was not identified?

3. (Page 6, line 159−163) The uncertainty of WSON estimated from propagating errors of WSTN, NO3− and NH4+ should be added.

4. (Page 6, line 175) The authors mentioned that [NH4+] in sea surface water typically less than 0.05 $\mu$mol L−1. Is this a common condition in the East China Sea and the

Northwestern Pacific Ocean during the sampling period (i.e., spring season)? Sea surface [NH4+] can vary depending on sampling season and locations.

References

Miyazaki, Y., Kawamura, K., Jung, J., Furutani, H. and Uematsu, M.: Latitudinal distributions of organic nitrogen and organic carbon in marine aerosols over the western North Pacific, Atmos. Chem. Phys., 11(7), 3037–3049, doi:10.5194/acp-11-3037-2011, 2011.

Nakamura, T., Ogawa, H., Maripi, D. K. and Uematsu, M.: Contribution of water soluble organic nitrogen to total nitrogen in marine aerosols over the East China Sea and western North Pacific, Atmos. Environ., 40(37), 7259–7264, doi:10.1016/j.atmosenv.2006.06.026, 2006.

---

## Author Comment (AC1) · 14 Jan 2018

This study presents measurements of concentrations of water-soluble total nitrogen (WSTN), nitrate ($NO_3^-$) and ammonium ($NH_4^+$), as well as the stable nitrogen isotopes of $\delta^{15}N$-WSTN and $\delta^{15}N$-$NO_3^-$ in marine aerosols sampled over the East China Seas (ECSs) and Northwest Pacific Ocean in spring 2014 and 2015. Dissolve organic nitrogen (DON) and $\delta^{15}N$-DON were also analyzed in surface sea water (SSW) in the spring of 2015. The highest concentrations of water-soluble nitrogen species were found in aerosol sampled in the ECSs, which suggests that anthropogenic emissions are a significant source of aerosol reactive nitrogen. The nitrogen species and isotopic compositions suggest that aerosol $\delta^{15}N$-WSTN values are mediated synergistically by the occupations of $NO_3^-$, $NH_4^+$ and water-soluble organic nitrogen (WSON) to WSTN

with their $\delta$15N in our observations. Meanwhile, our isotope mixing model indicates that DON in SSW is likely a source of primary WSON in aerosol, especially over the open ocean.
* * *

---

## Author Comment (AC3) · 14 Jan 2018

Point by point reply

This manuscript describes ship-board measurements of marine aerosols collected during two cruises around the East China Sea and the northwestern Pacific Ocean in 2014 and 2015. In this manuscript, authors reported concentrations of water-soluble total nitrogen (WSTN), water-soluble organic nitrogen (WSON), nitrate (NO3-) and ammonium (NH4+), and values of $\delta$15N-WSTN and $\delta$15N-NO3- in aerosols as well as dissolved organic nitrogen (DON) and NO3- concentrations and $\delta$15N-DON and $\delta$15N-NO3- values in sea surface water, which provide good indications where future studies can understand possible sources of atmospheric WSON and air-sea exchange of N species. I

believe that the contents, including data, of the manuscript should be eventually published because of scarcity of atmospheric WSON observation and its significance in biogeochemical N cycle.

Reply: Thanks for reviewer's appreciation of our data and the scientific significance.

I recommend publication of the manuscript after a major revision and the improvement of English.

Reply: We paid for editing service.

1. The title of this manuscript is "Sources of reactive nitrogen in marine aerosol over the Northwest Pacific Ocean in spring"; however, the authors mainly described the spatial distributions and concentrations of atmospheric reactive N species and potential sources of WSON. I recommend the authors to describe in their manuscript the sources of atmospheric inorganic N species, although they are relatively well-known compared to that of WSON.

Reply: Thanks for suggestion. We added more discussions of the sources of atmospheric inorganic N into Section 3.1 Paragraph 5.(Details can be found in supplement)

2. (Page 4, line 98-116) It is not clear that how many, what kind of aerosol samplers and filters were used during the cruises, and how avoided contamination from ship's exhaust. Was a impactor used to separate PM2.5 and PM10? More detail information on sampling method should be described in the manuscript, although the authors referred to Luo et al. (2016).

Reply: Aerosol sampling information including the instrument model, company, aerosol type and sampling filter have been added into Section 2.1. No separation for PM2.5 and PM10. Follow the reviewer's suggestion, the following explanation added into the Section 2.1. " To avoid self-contamination from the research vessel, the TSP sampler was installed on the top of the tower at the ship head, and aerosols were sampled only during travel. More information about self-contamination from ship exhaust can

be found in Luo et al. (2016)."

In general, a pre-combusted glass fiber or a quartz filter is used for determination of WSON. If the authors used the same method for aerosol sampling described in Luo et al. (2016), the authors should explain about the treatment of aerosol filter samples and field blank concentrations and blank correction, because Luo et al. (2016) used a Whatman 41 cellulose filter.

Reply: Descriptions of the blank were added into Section 2.2.1 to describe the field blanks and procedural blanks. " Eight filters of the same type as those used to collect samples were taken as blanks. All blank filters and aerosol samples were stored at −20 °C during the sampling periods and underwent the same extraction procedure. The $NO_3^-$, $NH_4^+$ and WSON content of the blank filters comprised less than 1%, 4% and 9%, respectively, of the average concentration of the corresponding N species in the aerosol samples."

For determination of DON in sea surface water, the authors mentioned that a 0.2 $\mu$m filter was used to remove particulate matters in sea surface water. Usually, a pre-combusted GF/F filter is used to remove particulate matter and minimize the influence of organic matter from the filters on DON concentration in seawater. Please update that what kind of filter was used for filtration of seawater sample. I am also wondering if any consensus reference material (CRM, e.g., deep Florida Strait water from Hansell lab, University of Miami) was used during DON measurement to check the accuracy of analysis.

Reply: The filter information has been added into the Section 2.1. The measured accuracy verified by our laboratory standard rather than other reference material, and the oxidation efficiency also has been added into the Section 2.2.3.(Details can be found in supplement)

(Page 6, line 188-Page 7, line 191) Dry deposition velocity. It is unclear if marine aerosols are segregated into PM2.5 and PM10 during the aerosol sampling as mentioned in question 2. Although size distributions of atmospheric N species can vary on meteorological conditions, it is known that, in the marine atmosphere, both atmospheric $NH_4^+$ and WSON primarily exist on fine mode aerosols, whereas atmospheric $NO_3^-$ is predominantly associated with coarse mode aerosols (e.g., Nakamura et al., 2006). I recommend the authors to describe more detail that what dry deposition velocity was used for each N species.

Reply: Thanks for the suggestion. The detail deposition velocity has been added into the Section 2.3. " The deposition velocities of water-soluble nitrogen species used herein were 2 cm s−1 for nitrate, 0.1 cm s−1 for ammonium, and 1.0 cm s−1 for WSON, which were consistent with our previous studies (Luo et al., 2016)."

4. (Page 8, line 228-232) The authors compared their $NH_4^+$ and $NO_3^-$ concentrations with those by Miyazaki et al. (2011) to explain why higher concentrations of inorganic N species were observed during the period of this study (spring). The authors mentioned that the study by Miyazaki et al. (2011) was carried out over the same regions. I doubt about it. The cruise by Miyazaki et al. (2011) was conducted from 44°N to 10°N along 155°E, which covers the subarctic to subtropical northwestern Pacific region. Although the study by Miyazaki et al. (2011) was carried out in summer, different sampling season is not the only reason why the authors observed high inorganic N species during their study period.

Reply: Agree. Since there were no more data of $NH_4^+$ and $NO_3^-$ in aerosol sampled on the same season and adjacent area, we can only compare with aerosol collected cover the western North Pacific Ocean. Base on the statistical significance ($p < 0.05$ for all cases), and we rewrite the Section 3.1 Paragraph 5 to discuss the variations of inorganic N in marine aerosol over the NWPO. (Details can be found in supplement)

5. (Page 8, line 233-242) The authors described that "Likely the source of WSON in background aerosol did not share the same source with $NH_4^+$ and $NO_3^-$" (line 234-235), as if DON in sea surface water is the only source of atmospheric WSON in

the open ocean. What is the grounds for this? Because high atmospheric WSON and inorganic N species concentrations were observed in the East China Sea and inorganic N was also detected in the open ocean, the long range transport of anthropogenic WSON to the open ocean should be considered.

Reply: Thanks for suggestion. We agree with that DON in sea surface water is not the only source of atmospheric WSON in the open ocean and aerosol WSON collected in the NWPO also influenced by the anthropogenic emission. Thus, we rewrite this sentence to make it read clearly, and the anthropogenic WSON to the open ocean also added in Section 3.1 Paragraph 6.(Details can be found in supplement)

6. (Page 8, line 252-Page 9, line 263) The results of characteristics of CHON molecular compounds shows that 13%, 3% and 19% of marine aerosols collected in the East China Sea, northwestern Pacific Ocean during dust period and northwestern Pacific Ocean during non dust period, were derived from biological sources, respectively. Does this mean that 87%, 97%, and 81% of marine aerosols collected in the same regions were affected by anthropogenic sources? It seems like the contribution of biogenic sources to atmospheric WSON is still low in the open ocean. What is the contribution of biologically-derived atmospheric WSON in the other oceanic regions?

Reply: Part of FT-ICRMS has been removed, the conclusion is not altered.

7. (Page 11, line 331-Page 12, line 355) The authors described that atmospheric reactive N dry deposition flux can account for 14%-58% of the low $_{15}N-NO_3^-$ in the northwestern Pacific Ocean during the spring. It is surprising to me that atmospheric reactive N deposition has a significant influence on $_{15}N-NO_3^-$ values. My question is that dry deposition of atmospheric reactive N is strong enough to affect or change $_{15}N-NO_3^-$ values below the thermocline in the northwestern Pacific Ocean? What is the depth of thermocline in the northwestern Pacific Ocean in the spring season? I recommend the authors to estimate the contribution of atmospheric reactive N dry deposition to primary production in their study area. I think most primary production

in the East China Sea and northwestern Pacific Ocean is controlled by nutrients in seawater, which implies that main factor for controlling _15N-NO3- values in the ocean is marine N cycle.

Reply: Thanks for the suggestion. We rewrite the Section 3.3 Paragraph 2.(Details can be found in supplement)

1. (Page 5, line 136-137) How did the authors obtain the recovery efficiency (i.e.,95-105% (n = 6)) of WSTN and TDN?

Reply: The recoveries of WSTN and TDN are the oxidation efficiency of prepared solution of N-containing organic and inorganic compounds standards (glycine, urea, ethylene diamine tetraacetic acid and ammonium sulfate) by the alkaline potassium persulfate. The following sentences had been added into the Section 2.2.3. "To verify the WSTN and TDN oxidation efficiency, N-containing organic and inorganic compound standards (specifically, glycine, urea, ethylene diamine tetraacetic acid, and ammonium sulphate) were prepared in solution at a concentration of 800 $\mu$M-N for oxidation analysis. The recoveries of the N-containing compound standards under oxidation by alkaline potassium persulfate were within $95 \sim 105\%$ (n = 6)"

2. (Page 5, 155-156) The authors mentioned that the extraction efficiency on a carbon basis was on average 46$\pm$24% (n = 44). Does it mean that 64% of organic compounds in the extract was not identified?

Reply: This part has been removed.

3. (Page 6, line 159-163) The uncertainty of WSON estimated from propagating errors of WSTN, NO3- and NH4+ should be added.

Reply: The errors propagation has been added into the Section 2.3. "The standard errors propagated through the WSON calculation for the 2014 data can be found in Luo et al. (2016). For 2015, the standard errors propagated through WSON calculation varied from sample to sample from 7 to 210%; the average standard error of all samples

was 33%."

(Page 6, line 175) The authors mentioned that [NH4+] in sea surface water typically less than 0.05 $\mu$mol L-1. Is this a common condition in the East China Sea and the Northwestern Pacific Ocean during the sampling period (i.e., spring season)? Sea surface [NH4+] can vary depending on sampling season and locations.

Reply: Reviewer is right. [NH4+] in sea surface water varies depending on sampling season and locations. However, NH4+ is much less than DON in this cruise agreeing with common sense for open oceans due to high bio-affinity of NH4+. In this version, we eliminate "typically" in old statement to avoid confusion. The revised statement is "Since the average [NH4+] in SSW at the selected sites during the 2015 cruise (12 sites and 23 samples) was 0.05 $\mu$M, which is much less than DON in $\mu$M level. . ."

Please also note the supplement to this comment:
https://www.atmos-chem-phys-discuss.net/acp-2017-846/acp-2017-846-AC3-supplement.zip

---

## Author Response (AR1)

**Point by point reply to Referee #1**

This paper presents a very interesting and important data set on Nr aerosol concentrations as well as the δ15N of total N and NO3- in aerosols. The aerosols were collected on ship transects from China to the Northwest Pacific Ocean, and as such are presented as marine aerosols. This is a dataset that should be published and it is important work.

Reply: Thanks for reviewer's appreciation of the merit of our work.

However, given the incredibly strong anthropogenic source strength of Nr in this region, I do not think these are representative of marine aerosols. Rather, this is a presentation of the impact of continental/anthropogenic aerosols on the coastal/near shore marine environment. It is a subtle, but important distinction.

Reply: Thanks for the suggestion. We added statements clearly present that the marine aerosol sampled in ECSs was co-influenced by both sea salt and continental/anthropogenic aerosols (Section 3.1 Paragraph 3). But in the main text, we still use "marine aerosol".

"The much higher water-soluble nitrogen species in the ECSs marine aerosol (compared to that in the NWPO aerosol) indicates that continental/anthropogenic Nr strongly affected the marine aerosol. However, the amounts of sea-salt ions (such as $Na^+$) in the ECSs aerosols sampled in both 2014 (123 $\pm$ 98 nmol m$^{-3}$; Luo et al., 2016) and 2015 (151 $\pm$164 nmol m$^{-3}$; Luo et al., unpublished data) were higher than those in land aerosol sampled during spring (23 $\pm$7.8 nmol m$^{-3}$ in Beijing; Zhang et al., 2013), which implies that those aerosols sampled in the ECSs were also significantly influenced by sea salt. Thus, we define the aerosol collected by ship over the ECSs as marine aerosol."

Section 2.2.5: The aerosol extracts do not contain salts, which is typically why SPE is used prior to FT-ICR MS analysis. It is not clear why this procedure was followed and some justification should be provided? It will lead to loss of organic carbon and organic nitrogen, indeed the % recoveries are < 50% and it does not seem appropriate. FT-ICR MS analysis using negative ion mode means it is not comparable to Altieri et al., or Wozniak et al. In those studies they analyzed samples in the positive ion mode. The negative ion mode would detect organonitrate compounds, whereas the amine and amino-acid compounds are detected in the positive ion mode. The listed elemental ratios based on positive ion mode analysis are not at all applicable in this work. Given the use of SPE and the negative ion mode analysis, the FT-ICRMS analysis needs to be removed from the manuscript, or interpreted in a completely different manner.

Reply: This comment is well taken. We removed all the text about the FT-ICRMS from the manuscript, and the conclusion is not altered.

I agree with page 1 line 24-26 – the anthropogenic signal is so strong that there is no way these aerosols represent a background signal at all. The classification needs to be explained in more detail, and changed to something more appropriate. It seems impossible that aerosols classified as "background" could have

such high concentrations of Nr. It is more likely that all of these aerosol samples are heavily influenced by anthropogenic pollution, with the signal declining as the polluted air is mixed with clean air off shore. Any local marine signal would be swamped by the large continental sources.

Reply: Agree with the reviewer. We defined the term "background" in the Section 2.1 Paragraph 2 prior to Discussion. "Hereafter, we define background aerosol as aerosol not impacted by either dust or sea fog,rather than representing pristine conditions, the background is an environmental baseline collected within the study area during the investigation period."

The similarity in 2014 and 2015 WSON concentrations is interesting given different chlorophyll fields. It is critical that some basic sampling information such as aerosol size is presented.

Reply: The sampling information has been added into the Section 2.1 Paragraph 1. "Total suspended particulate samples were collected using a high-volume sampler (TE-5170D; Tisch Environmental, Inc.) with Whatman®41 cellulose filters …"

In addition, the authors need to provide information on field blanks and procedural blanks, especially for WSON and ammonium concentrations.

Reply: Descriptions of the blank were added into the Section 2.2.1 to descript the field blanks and procedural blanks.

"Eight filters of the same type as those used to collect samples were taken as blanks. All blank filters and aerosol samples were stored at –20 ℃ during the sampling periods and underwent the same extraction procedure. The $NO_3^-$,$NH_4^+$ and WSON content of the blank filters comprised less than 1%, 4% and 9%, respectively, of the average concentration of the corresponding N species in the aerosol samples."

In Section 2.2.3, please explain the recoveries of WSTN and TDN? How does n=6 if there were 44 and 39 aerosol samples analyzed?

Reply: The recoveries of WSTN and TDN represent the oxidation efficiency of prepared solution of N-containing organic and inorganic compounds standards (glycine, urea, ethylene diamine tetraacetic acid and ammonium sulfate) by using the alkaline potassium persulfate.

The following sentences had been added into the Section 2.2.3.

"To verify the WSTN and TDN oxidation efficiency, N-containing organic and inorganic compound standards (specifically, glycine, urea, ethylene diamine tetraacetic acid, and ammonium sulphate) were prepared in solution at a concentration of 800 μM-N for oxidation analysis. The recoveries of the N-containing compound standards under oxidation by alkaline potassium persulfate were within 95 ~ 105%

(n = 6)."
* * *
What dry deposition velocities are used? There are large uncertainties associated with dry deposition velocities, which are size specific. If the aerosols are not size segregated, how can you apply a size-specific dry deposition velocity? These are going to be highly uncertain estimates and should be treated as such.

Reply: We agree. The dry deposition velocity varies by more than 3 orders of magnitude with particle size ranging from 0.1 to 100 μm (Hoppel et al., 2002). Thus, it is really hard to accurately estimate the Nr dry deposition by using TSP sample.

In general, ammonium appears in submicron mode from 0.1 to 1 μm, with a small fraction residing in the coarser mode, by contrast, nitrate is distributed mainly in supermicron size ranging from 1 to 10 μm while WSON appears in a wide size spectrum. Thus, for any water-soluble nitrogen species, using a fixed deposition velocity to calculate the dry deposition might cause under- or over- estimation. In our observation, wind speed ranged from 0.8 to 18m s$^{-1}$ under wide RH ranging from 40 to 100%. Thus, it is not possible to provide variable dry deposition velocities for estimation Nr dry deposition under a wide range of environmental conditions; assumptions were made based on existing knowledge. Here, deposition velocity of 2 cm s$^{-1}$ was applied for nitrate, 0.1 cm s$^{-1}$ for ammonium and 1.0 cm s$^{-1}$ for WSON. We have detailed the issue of size and deposition velocity of nitrate, ammonium and WSON in our previous ACP paper (Luo et al.,2016). Following the reviewer`s suggestion, the descriptions of deposition velocities of water-soluble nitrogen species were briefed in the Section 2.3.

"The deposition velocities of water-soluble nitrogen species used herein were 2 cm s$^{-1}$ for nitrate, 0.1 cm s$^{-1}$ for ammonium, and 1.0 cm s$^{-1}$ for WSON, which were consistent with our previous studies (Luo et al., 2016)."
* * *
Page 7 line 199-201: It's not clear what size the aerosols are, and so this comparison is difficult to understand. Regardless, these concentrations are incredibly high, and indicative of strong pollution sources.

Reply: Except the case for the Xi`an city, which was PM$_{10}$, all other aerosols were collected comparably by TSP. We explicitly described the difference in the revised version in the Section 3.1 Paragraph 2.
* * *
If the authors want to claim that any of these aerosols are "background" aerosols, they need to find other background sites that have such high Nr concentrations.

Reply: Replied above.

Also, the data are very difficult to see on the log scale, it's a very large range of concentrations to present on one figure. This should be separated somehow.

Reply: Thanks for this suggestion. In this version, we added one more figure (i.e., y-axis is presented in liner scale) into the supplementary material (Fig. S4).

Page 7 line222, you can't tell the difference between 2014 and 2015 in the figures due to the log scale. A different way of presenting the data would help.

Reply: The y-axis now is in liner scale (attached). We also added descriptions of statistical significance ($p < 0.05$ for all cases) into the figure 3 caption.

I agree with the authors conclusion on page 8 paragraph 2 – the "background" aerosols should be re-labeled as it is very misleading.

Reply: We added the definition of "background" in the Section 2.1 Paragraph 2.

Page 8 paragraph 3 – it is important to note that the surface seawater DON is the most likely source of primary WSON aerosols, but secondary WSON aerosols have many other sources, including e.g., surface ocean VOC emissions that go on to oxidize in the atmosphere and form secondary N-containing SOA.

Reply: Thanks for the suggestion. We added description of secondary WSON aerosols into the Section 3.1 Paragraph 6.

"However, the sources of marine aerosol WSON are complex mixture which composed of primary marine organic N and secondary N-containing organic aerosol. Biogenic organic material in SSW can be injected into the atmosphere to form an ice cloud via bubble bursting at the atmosphere-ocean interface (Wilson et al., 2015), this is probably the primary WSON aerosol source. Volatile organic compounds emitted from the surface ocean can react with $NO_x$ and $NH_x$ in the atmosphere to form secondary N-containing organic aerosol (Fischer et al., 2014; Liu et al., 2015). "

Section 3.2 should be removed or reinterpreted given the focus on negative mode CHON compounds identified here.

Reply: The part of FT-ICRMS has been removed.

Page 9 section 3.3 second paragraph. There is a large difference in WSTN _15N from 2014 to 2015 in these aerosols. This should be discussed. Figure 6 and discussion thereof: This is not a valid approach to

understanding what is driving the 15N of WSTN. A cross-plot of _15N-NO3- vs 15N-WSTON would provide more information on the influence of nitrate on the total N isotopic composition. Looking at figure 5, it looks like nitrate _15N is a main control on the _15N-WSTN. The lack of correlation between _15N WSTN and the relative concentration of NO3- is not useful. The relationship between the _15N RN and the relative concentrations of NH4+ would be useful, but is not presented.

Reply: Thanks for the suggestions. The scatter plots of $\delta^{15}$N-WSTN *vs.* $\delta^{15}$N-NO$_3^-$, $\delta^{15}$N-WSTN *vs.* NO$_3^-$, $\delta^{15}$N-RN *vs.* NH$_4^+$ and $\delta^{15}$N-RN *vs.* NH$_4^+$/RN have been added as Figure S7 and Figure S8 in the new version, and we redraw the Figure 5 and we rewrite the Section 3.2 Paragraph 1-3.

Page 10 paragraph line 300, it is also a possibility that the aerosol WSON is secondary organic aerosol, which may have had its 15N altered by transport or chemical reactions. This is a very over-simplified approach to the interpretation of the 15N-WSON data.

Reply: We agree with the reviewer, however, there are limited studies of $^{15}$N about the marine aerosol WSON. The multiple sources of marine aerosol WSON and $\delta^{15}$N -WSON fractionation during the processes of secondary N-containing organic aerosol formation are not clear to date. We discussed the possible causes to modulate $\delta^{15}$N-WSON in the last paragraph in Section 3.2 and highlighted more studies in future are needed for the secondary marine N-containing organic aerosol, particularly from the $^{15}$N scope.

"these high $\delta^{15}$N-RN values may be attributable to $\delta^{15}$N fractionation and $^{15}$N enrichment in the WSON during processes such as secondary N-containing organic aerosol formation by the reaction of NH$_x$ or NO$_x$ with organic aerosol (Fischer et al., 2014; Liu et al., 2015), complex atmospheric chemical reactions (i.e. the photolysis of organic nitrogen into ammonium; Paulot et al., 2015), aerosol WSON aging process, and in-cloud scavenging (Altieri et al., 2016). More studies are needed to explore nitrogen transformation processes, especially those focusing on secondary N-containing organic aerosol in the atmosphere from an isotopic perspective."

Section 3.4 is too speculative given the limited information presented. Are the ammonium, nitrate, and WSON concentrations statistically different from 2014 to 2015 and between the three classifications? Is there a statistically significant relationship between the _15N of DON in seawater or _15N NO3- in seawater and the _15N of TN, NO3- , or RN in the aerosols?

Reply: Thanks for the suggestion. The scatter plot of $\delta^{15}$N-DON in SSW vs. $\delta^{15}$N of aerosol NO$_3^-$, WSTN and RN in 2015 cruise are attached below. There were no significant relationship in scatter plots of the $\delta^{15}$N-DON in SSW against the aerosol $\delta^{15}$N-NO$_3^-$, RN and WSTN sampled correspondingly in time and space. According to the reviewer`s suggestion, we rewrite the Section 3.3 Paragraph 2.

Table 1. How are aerosol concentrations volume weighted? Is this a mass weighted average?

Reply: The volume weighted mean (C) calculated by the following equation:

$$C = \sum_{i=1}^{n} C_i V_i \ / \ \sum_{i=1}^{n} V_i$$

where $C_i$ is the concentration of water-soluble nitrogen species in aerosol, $V_i$ is the sampling volume for an aerosol, n is the number of sample.

Figure 1. It is not clear what "regional wind streamlines" are or where they came from. The blue on the background of the figure makes it difficult to see the symbols.

Reply: We enlarged streamlines and modified background color accordingly in the Figure 1.

Figure 3. The caption says a is nitrate and b is ammonium, but they are labeled in the opposite manner.

Reply: Corrected.

Figure 4 should be removed.

Reply: Removed.

Figure 6. The caption says a is concentration and b is _15N but the plots are the opposite.

Reply: Corrected.

Abstract: Line 14 insert "of" between transport and anthropogenic, line 15 "continents may exert a profound impact", line 16 should read "surface ocean" instead of "marine biogenic",

Reply: Modified as suggested.

line 18 do the authors mean in the open ocean or do they mean in the atmosphere?,

Reply: Modified as suggested.

Line 26 are the concentrations statistically higher in 2014? If so this should be presented in the text.

Reply: We presented the statistical significance ($p < 0.05$ for all cases) between 2014 and 2015 in this version.

Introduction: Define SSW on first use.

Reply: Defined in the Introduction.

Page 3 paragraph 1 should clearly state that they are referring to primary WSON aerosols.

Reply: Thanks. Primary WSON aerosol has been clearly stated in the new version.

Page 8 second paragraph. It's not clear what is meant by "atmospheric diffusion"

Reply: "Atmospheric diffusion" has been changed into "atmospheric long-range transport"

Altieri, K. E., Fawcett, S. E., Peters, A. J., Sigman, D. M., and Hastings, M. G.: Marine biogenic source of atmospheric organic nitrogen in the subtropical North Atlantic, Proceedings of the National

Academy of Sciences of the United States of America, 113, 925-930, doi:10.1073/pnas.1516847113, 2016.

Fischer, E. V., Jacob, D. J., Yantosca, R. M., Sulprizio, M. P., Millet, D. B., Mao, J., Paulot, F., Singh, H. B., Roiger, A., Ries, L., Talbot, R. W., Dzepina, K., and Pandey Deolal, S.: Atmospheric peroxyacetyl nitrate (PAN): a global budget and

source attribution, Atmospheric Chemistry and Physics, 14, 2679-2698, doi:10.5194/acp -14-2679-2014, 2014.

Hoppel, W., Frick, G., and Fitzgerald, J.: Surface source function for sea-salt aerosol and aerosol dry deposition to the ocean surface, J. Geophys. Res.-Atmos., 107, AAC7.1–AAC7.17,doi:10.1029/2001J D002014, 2002.

Liu Y, Liggio J, Staebler R, et al. Reactive uptake of ammonia to secondary organic aerosols: kinetics of organonitrogen formation[J]. Atmospheric Chemistry & Physics, 2015, 15(23):17449-17490.

Luo, L., Yao, X. H., Gao, H. W., Hsu, S. C., Li, J. W., and Kao, S. J.: Nitrogen speciation in various types of aerosols in spring over the northwestern Pacific Ocean. Atmospheric Chemistry and Physics, 16, 325-341, doi:10.5194/acp-16-325-2016, 2016.

Paulot, F., Jacob, D. J., Johnson, M. T., Bell, T. G., Baker, A. R., Keene, W. C., Lima, I. D., Doney, S. C., and Stock, C. A.: Global oceanic emission of ammonia: Constraints from seawater and atmospheric observations, Global Biogeochemical Cycles, 29, 1165-1178, doi:10.1002/2015gb 005106, 2015.

Wilson, T. W., Ladino, L. A., Alpert, P. A., Breckels, M. N., Brooks, I. M., Browse, J., Burrows, S. M., Carslaw, K. S., Huffman, J. A., Judd, C., Kilthau, W. P., Mason, R. H., McFiggans, G., Miller, L. A., Najera, J. J., Polishchuk, E., Rae, S., Schiller, C. L., Si, M., Temprado, J. V., Whale, T. F., Wong, J. P., Wurl, O., Yakobi-Hancock, J. D., Abbatt, J. P., Aller, J. Y., Bertram, A. K., Knopf, D. A., and Murray, B. J.: A marine biogenic source of atmospheric ice-nucleating particles, Nature, 525, 234-238, doi:10.1038/nature14986, 2015.

Zhang, R., Jing, J., Tao, J., Hsu, S.-C., Wang, G., Cao, J., Lee, C. S. L., Zhu, L., Chen, Z., Zhao, Y., and Shen, Z.: Chemical characterization and source apportionment of PM2.5 in Beijing: seasonal perspective, Atmos. Chem. Phys., 13, 7053–7074,doi:10.5194/acp-13-7053-2013, 2013.

**Point by point reply to Referee#2**

This manuscript describes ship-board measurements of marine aerosols collected during two cruises around the East China Sea and the northwestern Pacific Ocean in 2014 and 2015. In this manuscript, authors reported concentrations of water-soluble total nitrogen (WSTN), water-soluble organic nitrogen (WSON), nitrate (NO3-) and ammonium (NH4+), and values of δ15N-WSTN and δ15N-NO3- in aerosols as well as dissolved organic nitrogen (DON) and NO3- concentrations and δ15N-DON and δ15N-NO3- values in sea surface water, which provide good indications where future studies can understand possible sources of atmospheric WSON and air-sea exchange of N species. I believe that the contents, including data, of the manuscript should be eventually published because of scarcity of atmospheric WSON observation and its significance in biogeochemical N cycle.

Reply: Thanks for reviewer's appreciation of our data and the scientific significance.

I recommend publication of the manuscript after a major revision and the improvement of English.

Reply: We paid for editing service.

1. The title of this manuscript is "Sources of reactive nitrogen in marine aerosol over the Northwest Pacific Ocean in spring"; however, the authors mainly described the spatial distributions and concentrations of atmospheric reactive N species and potential sources of WSON. I recommend the authors to describe in their manuscript the sources of atmospheric inorganic N species, although they are relatively well-known compared to that of WSON.

Reply: Thanks for suggestion. We added more discussions of the sources of atmospheric inorganic N into the Section 3.1 Paragraph 5.

2. (Page 4, line 98-116) It is not clear that how many, what kind of aerosol samplers and filters were used during the cruises, and how avoided contamination from ship's exhaust. Was a impactor used to separate PM2.5 and PM10? More detail information on sampling method should be described in the manuscript, although the authors referred to Luo et al. (2016).

Reply: Aerosol sampling information including the instrument model, company, aerosol type and sampling filter have been added into the Section 2.1.

No separation for PM2.5 and PM10.

Follow the reviewer's suggestion, the following explanation added into the Section 2.1. " To avoid self-contamination from the research vessel, the TSP sampler was installed on the top of the tower at the ship head, and aerosols were sampled only during travel. More information about self-contamination from ship exhaust can be found in Luo et al. (2016)."

In general, a pre-combusted glass fiber or a quartz filter is used for determination of WSON. If the authors used the same method for aerosol sampling described in Luo et al. (2016), the authors should explain about the treatment of aerosol filter samples and field blank concentrations and blank correction, because Luo et al. (2016) used a Whatman 41 cellulose filter.

Reply: Descriptions of the blank were added into the Section 2.2.1 to describe the field blanks and procedural blanks.

" Eight filters of the same type as those used to collect samples were taken as blanks. All blank filters and aerosol samples were stored at –20 ℃ during the sampling periods and underwent the same extraction procedure. The $NO_3^-$, $NH_4^+$ and WSON content of the blank filters comprised less than 1%, 4% and 9%, respectively, of the average concentration of the corresponding N species in the aerosol samples."

For determination of DON in sea surface water, the authors mentioned that a 0.2 μm filter was used to remove particulate matters in sea surface water. Usually, a pre-combusted GF/F filter is used to remove particulate matter and minimize the influence of organic matter from the filters on DON concentration in seawater. Please update that what kind of filter was used for filtration of seawater sample. I am also wondering if any consensus reference material (CRM, e.g., deep Florida Strait water from Hansell lab, University of Miami) was used during DON measurement to check the accuracy of analysis.

Reply: The filter information has been added into the Section 2.1.

The measured accuracy verified by our laboratory standard rather than other reference material, and the oxidation efficiency also has been added into the Section 2.2.3.

(Page 6, line 188-Page 7, line 191) Dry deposition velocity. It is unclear if marine aerosols are segregated into PM2.5 and PM10 during the aerosol sampling as mentioned in question 2. Although size distributions of atmospheric N species can vary on meteorological conditions, it is known that, in the marine atmosphere, both atmospheric NH4+ and WSON primarily exist on fine mode aerosols, whereas atmospheric NO3- is predominantly associated with coarse mode aerosols (e.g., Nakamura et al., 2006). I recommend the authors to describe more detail that what dry deposition velocity was used for each N species.

Reply: Thanks for the suggestion. The detail deposition velocity has been added into the Section 2.3.

" The deposition velocities of water-soluble nitrogen species used herein were 2 cm s$^{-1}$ for nitrate, 0.1 cm s$^{-1}$ for ammonium, and 1.0 cm s$^{-1}$ for WSON, which were consistent with our previous studies (Luo et al., 2016)."

4. (Page 8, line 228-232) The authors compared their NH4+ and NO3- concentrations with those by Miyazaki et al. (2011) to explain why higher concentrations of inorganic N species were observed during the period of this study (spring). The authors mentioned that the study by Miyazaki et al. (2011) was carried out over the same regions. I doubt about it. The cruise by Miyazaki et al. (2011) was conducted from 44 °N to 10 °N along 155 °E, which covers the subarctic to subtropical northwestern Pacific region.

Although the study by Miyazaki et al. (2011) was carried out in summer, different sampling season is not the only reason why the authors observed high inorganic N species during their study period.

Reply: Agree. Since there were no more data of NH4+ and NO3- in aerosol sampled on the same season and adjacent area, we can only compare with aerosol collected cover the western North Pacific Ocean. Base on the statistical significance (p < 0.05 for all cases), and we rewrite the Section 3.1 Paragraph 5 to discuss the variations of inorganic N in marine aerosol over the NWPO.

5. (Page 8, line 233-242) The authors described that "Likely the source of WSON in background aerosol did not share the same source with NH4+ and NO3-" (line 234-235), as if DON in sea surface water is the only source of atmospheric WSON in the open ocean. What is the grounds for this? Because high atmospheric WSON and inorganic N species concentrations were observed in the East China Sea and inorganic N was also detected in the open ocean, the long range transport of anthropogenic WSON to the open ocean should be considered.

Reply: Thanks for suggestion. We agree with that DON in sea surface water is not the only source of atmospheric WSON in the open ocean and aerosol WSON collected in the NWPO also influenced by the anthropogenic emission. Thus, we rewrite this sentence to make it read clearly, and the anthropogenic WSON to the open ocean also added in the Section 3.1 Paragraph 6.

6. (Page 8, line 252-Page 9, line 263) The results of characteristics of CHON molecular compounds shows that 13%, 3% and 19% of marine aerosols collected in the East China Sea, northwestern Pacific Ocean during dust period and northwestern Pacific Ocean during non dust period, were derived from biological sources, respectively. Does this mean that 87%, 97%, and 81% of marine aerosols collected in the same regions were affected by anthropogenic sources? It seems like the contribution of biogenic sources to atmospheric WSON is still low in the open ocean. What is the contribution of biologically-derived atmospheric WSON in the other oceanic regions?

Reply: Part of FT-ICRMS has been removed, the conclusion is not altered.

7. (Page 11, line 331-Page 12, line 355) The authors described that atmospheric reactive N dry deposition flux can account for 14%-58% of the low _15N-NO3□ in the northwestern Pacific Ocean during the spring. It is surprising to me that atmospheric reactive N deposition has a significant influence on _15N-NO3□ values. My question is that dry deposition of atmospheric reactive N is strong enough to affect or change _15N-NO3□ values below the thermocline in the northwestern Pacific Ocean? What is the depth of thermocline in the northwestern Pacific Ocean in the spring season? I recommend the authors to estimate the contribution of atmospheric reactive N dry deposition to primary production in their study area. I think most primary production in the East China Sea and northwestern Pacific Ocean is controlled by nutrients in seawater, which implies that main factor for controlling _15N-NO3□ values in the ocean is marine N cycle.

Reply: Thanks for the suggestion. We rewrite the Section 3.3 Paragraph 2.

1. (Page 5, line 136-137) How did the authors obtain the recovery efficiency (i.e.,95-105% (n = 6)) of WSTN and TDN?

Reply: The recoveries of WSTN and TDN are the oxidation efficiency of prepared solution of N-containing organic and inorganic compounds standards (glycine, urea, ethylene diamine tetraacetic acid and ammonium sulfate) by the alkaline potassium persulfate.

The following sentences had been added into the Section 2.2.3.

"To verify the WSTN and TDN oxidation efficiency, N-containing organic and inorganic compound standards (specifically, glycine, urea, ethylene diamine tetraacetic acid, and ammonium sulphate) were prepared in solution at a concentration of 800 μM-N for oxidation analysis. The recoveries of the N-containing compound standards under oxidation by alkaline potassium persulfate were within 95 ~ 105% (n = 6)"

2. (Page 5, 155-156) The authors mentioned that the extraction efficiency on a carbon basis was on average $46\pm24\%$ (n = 44). Does it mean that 64% of organic compounds in the extract was not identified?

Reply: This part has been removed.

3. (Page 6, line 159-163) The uncertainty of WSON estimated from propagating errors of WSTN, NO3- and NH4+ should be added.

Reply: The errors propagation has been added into the Section 2.3.

"The standard errors propagated through the WSON calculation for the 2014 data can be found in Luo et al. (2016). For 2015, the standard errors propagated through WSON calculation varied from sample to sample from 7 to 210%; the average standard error of all samples was 33%."

(Page 6, line 175) The authors mentioned that [NH4+] in sea surface water typically less than 0.05 μmol L-1. Is this a common condition in the East China Sea and the Northwestern Pacific Ocean during the sampling period (i.e., spring season)? Sea surface [NH4+] can vary depending on sampling season and locations.

Reply: Reviewer is right. [NH4+] in sea surface water varies depending on sampling season and locations. However, NH4+ is much less than DON in this cruise agreeing with common sense for open oceans due to high bio-affinity of NH4+.

In this version, we eliminate "typically" in old statement to avoid confusion. The revised statement is "Since the average [$NH_4^+$] in SSW at the selected sites during the 2015 cruise (12 sites and 23 samples) was 0.05 μM, which is much less than DON in μM level…"

**List of all the relevant changes made in the manuscript**

1, Aerosol sampling information including the instrument model, company, aerosol type and sampling filter have been added into the Section 2.1 Paragraph 1.

2, The following sentences were added into the Section 2.1 Paragraph 1. " To avoid self-contamination from the research vessel, the TSP sampler was installed on the top of the tower at the ship head, and aerosols were sampled only during travel. More information about self-contamination from ship exhaust can be found in Luo et al. (2016)."

3, We define the term "background" in the Section 2.1 Paragraph 2.

4, Descriptions of the blank were added into the Section 2.2.1 to describe the field blanks and procedural blanks.

5, The "oxidation efficiency" also has been added into the Section 2.2.3.

6, The errors propagation of WSON calculation has been added into the Section 2.3 Paragraph 1.

7, We redescribe the [NH4+] in sea surface water in the Section 2.3 Paragraph 3.

8, The detail deposition velocity has been added into the Section 2.3 Paragraph 4.

9, We explicitly describe the difference of aerosol size in the Section 3.1 Paragraph 2.

10, We add statements clearly present that the marine aerosol sampled in ECSs was co-influenced by both sea salt and continental/anthropogenic aerosols in the Section 3.1 Paragraph 3.

11, We rewrite the Section 3.1 Paragraph 5.

12, We rewrite the Section 3.1 Paragraph 6.

13, Part of FT-ICRMS has been removed from the revised manuscript.

14, We rewrite the Section 3.2 Paragraph 1-3.

15, We rewrite the Section 3.2 Paragraph 6.

16, We rewrite the Section 3.3 Paragraph 2.

17, We modified the Figure 1.

18, We modified the Figure 3.

19, We redraw the Figure 5.

20, Figure S4, Figure S7, Figure S8 were added into the supplement.

21, Other minor changes have been done.

22, we paid for editing service.

[revised manuscript text omitted]

---

## Referee Report (RR1)

Re-review of Luo et al. submission:

The authors have addressed a substantial amount of both reviewers concerns. However, there are a few points that I think are critical that have not been adequately addressed. I still think it is very misleading to call any of these background aerosols. The authors are using this as a designation for aerosols that are not influenced by sea fog or dust, but there is a history in the literature of what a background aerosol is, and this is not an appropriate usage, even with a definition in the manuscript.

It appears based on the authors reply that there are no true experimental blanks. They measured the values on the filters alone, i.e., a filter blank, but they never deployed those filters into their setup on the cruise to determine a methodological blank. This is a serious problem as these types of collections on ship can quite easily have handling blanks due to ship exhaust and not being in a clean laboratory.

The average concentration for the aerosols is not useful given the incredibly wide range in aerosol concentrations reported. In addition, the filter blank value should be used to correct the reported values.

Ammonium is normally fine mode aerosol, except in coastal areas with mixed pollution and marine aerosols, where it frequently is present in the coarse mode (e.g., Yeatman, S.G., Spokes, L.J., Dennis, P.F. & Jickells, T.D. 2001. Comparisons of aerosol nitrogen isotopic composition at two polluted coastal sites.). I do not think it is appropriate to use deposition velocities for TSP samples given the potential for fine mode ammonium nitrate due to high concentrations pollutants (i.e., nitrate in the fine mode), and coarse mode ammonium. Everything related to the dry deposition N fluxes should be removed. Knowing the actual size of each aerosol type in a complex highly polluted yet marine area is critical, it is not okay to just use the typical assumptions. All of section 3.3 should be removed as well as Figure 8.

The propagated errors for WSON concentrations are huge – this needs to be visible in all figures and its impact on interpretation should be accounted for.

---

## Author Response (AR2)

**Point-by-point reply to the Referee**

Re-review of Luo et al. submission:

The authors have addressed a substantial amount of both reviewers concerns. However, there are a few points that I think are critical that have not been adequately addressed. I still think it is very misleading to call any of these background aerosols. The authors are using this as a designation for aerosols that are not influenced by sea fog or dust, but there is a history in the literature of what a background aerosol is, and this is not an appropriate usage, even with a definition in the manuscript.

**Reply:** Thank you for your suggestion. 'Background aerosol' has been widely used to describe the aerosol collected in a clear site relative to highly polluted areas (Allan et al., 2006; Niemi et al., 2006; Arndt et al., 2017). Following previous definitions of 'background aerosol', we changed the relevant sentences to 'Compared with the ECSs, which were strongly influenced by anthropogenic emissions, the NWPO (open ocean) was relatively clear. Hereafter, we define "background aerosol" as aerosol collected in the NWPO without influence from dust and sea fog during the investigation period.'

It appears based on the authors reply that there are no true experimental blanks. They measured the values on the filters alone, i.e., a filter blank, but they never deployed those filters into their setup on the cruise to determine a methodological blank. This is a serious problem as these types of collections on ship can quite easily have handling blanks due to ship exhaust and not being in a clean laboratory.

**Reply:** Thank you for pointing this out; it is our mistake. We did not state the process of measuring blanks clearly. We rewrote the relevant procedures in Section 2.2.1:

'A total of eight filters of the same type as those used to collect samples were taken as blanks. Before storage, the blank filter was placed on the filter holder, then the filter holder with the blank filter was subsequently installed in the TSP sampler on the top of the ship under the vacuum motor power-off for 5 min, after which the blank filter was retrieved. All blank filters and aerosol samples were stored at –20 ℃ during the sampling periods and underwent the same extraction procedures.'

The average concentration for the aerosols is not useful given the incredibly wide range in aerosol concentrations reported.

**Reply:** Referee is right. The range of $N_r$ concentrations in aerosol was large, ranging by a factor of 40, which was large but within the ranges of previous studies (from ~ 20 to > 100) for marine aerosol on the coast, marginal sea and open ocean (Yeatman et al., 2001; Nakamura et al., 2005; Morin et al., 2009; Miyazaki et al., 2011; Zhu et al., 2013; Violaki et al., 2015), due to meteorological conditions (such as sea fog, rain and dust storm), the sources of air masses (such as anthropogenic sources) and the distance from land. Compared with other studies, the upper and lower bounds of our data, averages and standard deviation have been listed in Table 1. The significant differences at the $p < 0.05$ level between different years are shown in Figure 3.

In addition, the filter blank value should be used to correct the reported values.

**Reply:** The reported data have been corrected by subtracting the filter blank. To clarify this, we added the following sentence to Section 2.2.1: 'The data presented here have been corrected for blanks.'

Ammonium is normally fine mode aerosol, except in coastal areas with mixed pollution and marine aerosols, where it frequently is present in the coarse mode (e.g., Yeatman, S.G., Spokes, L.J., Dennis, P.F. & Jickells, T.D. 2001. Comparisons of aerosol nitrogen isotopic composition at two polluted coastal sites.). I do not think it is appropriate to use deposition velocities for TSP samples given the potential for fine mode ammonium nitrate due to high concentrations pollutants (i.e., nitrate in the fine mode), and coarse mode ammonium. Everything related to the dry deposition N fluxes should be removed. Knowing the actual size of each aerosol type in a complex highly polluted yet marine area is critical, it is not okay to just use the typical assumptions. All of section 3.3 should be removed as well as Figure 8.

**Reply:** While we appreciate the reviewer's comment, we disagree with it. The deposition velocity is not only affected by the aerosol size distribution, but is also regulated by meteorological conditions (Duce et al., 1991; Hoppel et al., 2002) and underlying surface (Piskunov, 2009). To clearly communicate our assumptions and the potential bias, more descriptions about the deposition velocity have been added to Section 2.3 Paragraph 4. In addition, to promote the value of the rare

dataset, the reasonable estimation under the explicit assumption should be acceptable with uncertainties. Thus, we preferred to keep the following paragraph:

'In addition to particle size, $V_i$ was controlled by the meteorological conditions (wind speed and relative humidity; Duce et al., 1991; Hoppel et al., 2002) and underlying surface (smooth or rough; Piskunov, 2009). The $V_i$ simulated by the model varied from 0.01 to 10 cm s$^{-1}$ under wind speeds from 5 to 30 m s$^{-1}$ and particle size from 0.1 to 100 μm (Hoppel et al., 2002). In our observations, wind speed ranged from 0.1 to 18 m s$^{-1}$, with relative humidity ranging from 40 to 100% (for 2014 data, see Luo et al. (2016) and for 2015 data see Fig. S2), which was variable, thus prohibiting an accurate deposition estimate. Moreover, we did not obtain information about the size distribution. Previous studies showed that most of the $NO_3^-$ is distributed in supermicron size (ranging from 1 to 10 μm) in marine aerosol, with a small fraction in submicron size (ranging from 0.1 to 1 μm). On the other hand, $NH_4^+$ is mainly distributed in submicron size and only partly in supermicron size (Nakamura et al., 2005; Baker et al., 2010; Jung et al., 2013), except in coastal areas with mixed pollution and marine aerosols, where $NH_4^+$ was present in the coarse mode and $NO_3^-$ in the fine mode (Yeatman et al., 2001). In our case, the weather conditions, such as fog and dust, further affected the size distribution of aerosol $N_r$ (Mori et al., 2003; Yao and Zhang, 2012; Hsu et al., 2014). Therefore, in this study, the deposition velocities of water-soluble nitrogen species were set to 2 cm s$^{-1}$ for $NO_3^-$, 0.1 cm s$^{-1}$ for $NH_4^+$ and 1.0 cm s$^{-1}$ for WSON, which have also been widely used to estimate the marine aerosol $N_r$ dry deposition (Nakamura et al., 2005; Jung et al., 2013; Luo et al., 2016). However, bearing in mind, that any use of fixed deposition velocities to calculate the depositional flux of aerosol $N_r$ may cause under- or over-estimation.'

The propagated errors for WSON concentrations are huge – this needs to be visible in all figures and its impact on interpretation should be accounted for.

**Reply:** The uncertainties (~ 20 to > 100%) caused by error propagation during the calculation of WSON have been thoroughly discussed in previous studies (Mace and Duce, 2002; Cornell et al., 2003; Cape et al., 2011; Lesworth et al., 2010; Zamora et al., 2011). However, no error values were provided. Following the referee's suggestion, the error bar for WSON concentration has been made visible in corresponding figures and the error has been taken into account in associated WSON

discussions. We also rewrote the Section 2.3 Paragraph 1 to clarify the uncertainties of calculated WSON:

'The calculated WSON was subject to relatively large and variable uncertainties by error propagation, since $[NO_3^-]$ and $[NH_4^+]$ were generally much higher than WSON concentrations in most observations. Such error propagation was a common and unavoidable problem (Mace and Duce, 2002; Cornell et al., 2003; Cape et al., 2011; Lesworth et al., 2010; Zamora et al., 2011). In previous studies, data points with high relative uncertainties were excluded (>100%, Lesworth et al., 2010; Zamora et al., 2011) and the negative values were taken as zero (Mace and Duce, 2002; Cornell et al., 2003; Violaki et al., 2015). Following previous studies, we excluded both the negative data and those with high relative uncertainties to reduce the uncertainty of mean WSON.'

[revised manuscript text omitted]